# STEPWISE HIGH-LEVEL SEMANTIC ALIGNMENT FOR TEST-TIME ADAPTATION

## ABSTRACT

Test-Time Adaptation (TTA) aims to adapt a source-trained model to a target domain without access to source data or target labels. Among existing approaches, Source Distribution Estimation (SDE) is valued for its ability to preserve source discriminability and ensure stable adaptation. However, most SDE-based methods rely on aligning low-level features statistics like batch normalization, often resulting in class confusion and unstable decision boundaries under large domain shifts. To address this, we propose **SHLSA**, a *Stepwise High-Level Semantic Alignment* framework that incorporates semantic priors to align features in the high-level space, preserving category structure and enabling more stable, semantically consistent adaptation. Specifically, SHLSA introduces the *pseudo-source domain* as a semantic bridge between the source and target domains, enabling a more stable and effective stepwise domain alignment (**SDA**) from reliable to ambiguous regions. To further enhance semantic feature quality, we design a hierarchical feature aggregation (**HFA**) module that integrates local and global representations via attention, improving local consistency and global convergence. Building on these enriched features, we introduce a confidence-aware complementary learning (**CACL**) strategy to refine EMA-updated pseudo-labels by suppressing noise and improving semantic reliability, thereby enhancing supervision for target domain samples. Extensive experiments on standard TTA benchmarks demonstrate the superior performance and generalizability of SHLSA.

## 1 INSTRUCTION

Most machine learning methods assume that training and testing data are independently and identically distributed (i.i.d.) Liang et al. (2025). However, this assumption often fails in real-world scenarios due to limited data coverage and distribution shifts between source and target domains. To address such shifts, *Domain Generalization* (DG) learns domain-invariant features from labeled sources Zhou et al. (2022), while *Domain Adaptation* (DA) transfers knowledge from labeled source data to unlabeled target data Farahani et al. (2021). A more challenging setting, *Test-Time Adaptation* (TTA), adapts a source-trained model using only unlabeled target data, without access to source data.

To tackle this source-free, label-free scenario, TTA methods employ strategies such as pseudo labeling, consistency regularization, clustering, and SDE Liang et al. (2025). Among them, SDE stands out for its ability to preserve source discriminability and ensure stable adaptation under distribution shifts. It reconstructs an estimate of the source distribution through data generation, domain translation, or memory selection, and aligns it with the target domain via semi-supervised or adversarial learning. This process mitigates confirmation bias and promotes more reliable adaptation by explicitly retaining the structural priors of the source model.

However, existing SDE-based methods tend to focus on low-level feature alignment, such as matching textures or BatchNorm statistics Liang et al. (2020); Ding et al. (2022); Yang et al. (2021); Wang et al. (2022b). While effective in reducing shallow appearance gaps, these approaches often lead to class confusion and unstable decision boundaries under significant domain shifts due to the lack of semantic-level alignment. For example, SHOT approximates the source distribution using BatchNorm statistics, which can become unreliable under class imbalance or large-scale distributional changes Liang et al. (2020). In contrast, aligning features in high-level semantic space

helps maintain category structure and offers stronger invariance to low-level variations like texture, illumination, background, and scale. This semantic-aware alignment addresses class confusion and unstable boundaries caused by shallow feature alignment, providing a more robust and interpretable foundation for distribution matching and decision refinement under complex domain shifts.

Building on these observations, SHLSA is designed to facilitate robust TTA by decomposing conventional one-step alignment into two phased sub-alignments, enabling stepwise semantic transfer. The process begins by estimating domain divergence via self-entropy and partitioning the target domain accordingly. High-confidence target samples are selected to construct a *pseudo-source domain*, serving as a semantic bridge between the source and the remaining target data. SHLSA aligns the source and pseudo-source domains using universal semantic features extracted from an ImageNet-pretrained model Hoyer et al. (2023), which offer strong semantic priors and cross-domain generalization without requiring access to source data. Enriched with source semantics, the pseudo-source then guides the adaptation of lower-confidence target samples through semi-supervised learning, providing stable supervision via pseudo labels. To further enhance semantic representation under domain shift, we introduce a *hierarchical feature aggregation module* that integrates local and global representations via attention, improving object coverage and structural coherence. Finally, a *confidence-aware complementary learning strategy* refines pseudo labels by partitioning them into positive and negative components based on relative confidence, effectively suppressing semantic noise and enhancing pseudo label reliability. The main contributions are summarized as follows:

- We propose SHLSA, a stepwise feature alignment framework for TTA, which decomposes one-step alignment into two structured stages and introduces a pseudo-source domain as a semantic bridge, enabling effective adaptation from reliable to ambiguous regions.

- We propose a hierarchical feature aggregation module to enhance semantic representations by integrating local and global contexts, and a confidence-aware complementary learning strategy to suppress semantic noise and improve pseudo-label reliability.

## 2 RELATED WORK

**Source Distribution Estimation for TTA.** In the absence of source data, traditional domain adaptation (DA) and generalization (DG) methods become inapplicable. Recent TTA techniques address this by relying on the pretrained source model during inference. A key approach, SDE, treats TTA as a DA problem by constructing a pseudo-source domain from target data Liang et al. (2025). Techniques like adversarial training Nayak et al. (2021), style transfer Hu et al. (2022), and uncertainty-based sampling Liang et al. (2021) aim to approximate the source distribution. Virtual domain alignment methods, utilizing semi-supervised learning, adversarial training, or contrastive objectives, further align the pseudo-source and target domains Liang et al. (2021); Ding et al. (2023); Kurmi et al. (2021). These methods promote semantic consistency across domains. However, aligning high-level semantic features across domains remains a challenge, particularly under domain shifts. To address this, we propose a stepwise high-level semantic alignment framework, improving feature consistency from reliable to ambiguous regions. By introducing structured semantic guidance, our approach enhances cross-domain consistency and improves object-level completeness, leading to more reliable feature representations under domain shift.

**Pseudo Labeling for TTA.** A mainstream direction in TTA focuses on improving pseudo-label quality through denoising or weighting, with the goal of providing reliable supervision under source–target domain shifts Liang et al. (2020); Wang et al. (2024); Qu et al. (2022). In SDE, pseudo-labels are primarily used as auxiliary tools to approximate the source distribution Ilse et al. (2020). Common strategies include confidence-based filtering Wang et al. (2024), iterative self-training Hu et al. (2025), generative refinement Ding et al. (2023), and clustering-based assignment Liang et al. (2020); Qu et al. (2022). Most methods treat pseudo-labels as one-hot targets, which limits their semantic richness. Confidence-aware approaches from semi-supervised learning help mitigate this by better handling uncertain predictions and improving label quality Feng et al. (2024); Wang et al. (2022c). Meanwhile, recent work has demonstrated the effectiveness of hierarchical feature aggregation in capturing both local details and global context Hoyer et al. (2022b). In SHLSA, this design facilitates structured semantic fusion, leading to more complete object representations and more reliable pseudo-label generation.

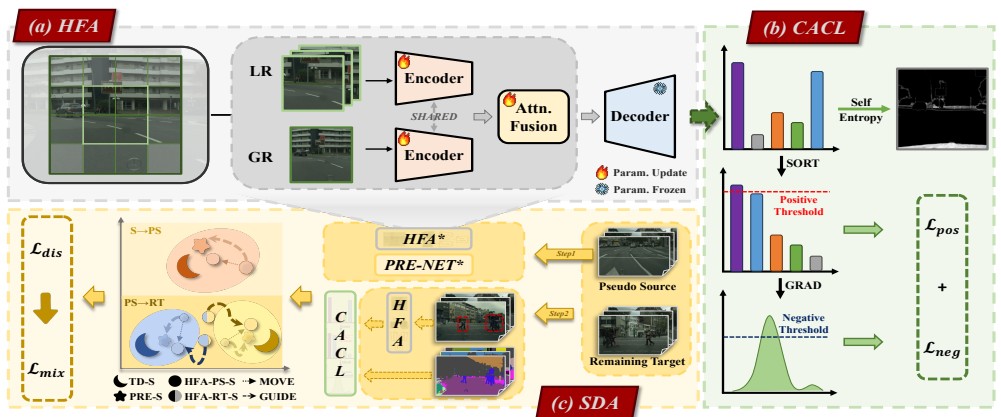

Figure 1: Overview of SHLSA, including: **(a) HFA**, **(b) CACL**, and **(c) SDA**. HFA* and PRE-NET* are their respective feature extractors. TD-S and PRE-S refer to the **s**emantic features derived from the **t**arget **d**omain (serving as the idealized semantics) and the **pre**-trained network, respectively. HFA-PS-S and HFA-RT-S correspond to the semantic features extracted from $D_{ps}$ and $D_{rt}$ via HFA. **Pseudo-code** is provided in Appendix A.1

## 3 PROPOSED METHOD

### 3.1 PROBLEM DEFINITION

Given a model $\mathcal{M}_s$ pre-trained on the source domain $D_s = \{\mathcal{X}_s, \mathcal{Y}_s\} = \{\boldsymbol{x}_s^i, y_s^i\}_{i=1}^{n_s}$, and unlabeled target domain data $D_t = \{\mathcal{X}_t\} = \{\boldsymbol{x}_t^i\}_{i=1}^{n_t}$, where $n_s$ and $n_t$ denote the number of samples in the source and target domains respectively, TTA aims to fine-tune the source model $\mathcal{M}_s$ using $\mathcal{D}_t$ to obtain an adapted model $\mathcal{M}_t$ that can effectively handle domain shifts from the source to the target domain. To prevent ambiguity in the context of pseudo-source domain construction, the target domain is further partitioned in this work as follows:

$$D_t = D_{ps} + D_{rt} = \{\{\mathcal{X}_{ps}\}, \{\mathcal{X}_{rt}\}\} = \{\{\boldsymbol{x}_{ps}^i\}_{i=1}^{n_{ps}}, \{\boldsymbol{x}_{rt}^j\}_{j=1}^{n_{rt}}\}, \tag{1}$$

where $D_{ps}$ and $D_{rt}$ denote the *pseudo-source domain* samples and the *remaining target domain* data, respectively. These subsets satisfy the conditions $D_{ps} \cap D_{rt} = \emptyset$, $D_{ps} \cup D_{rt} = D_t$, and $n_{ps} + n_{rt} = n_t$. The pseudo-source domain $D_{ps}$ is constructed by partitioning $D_t$ based on sample self-entropy, ensuring both $D_{ps}$ and $D_{rt}$ are disjoint subsets of the target domain.

### 3.2 OVERVIEW

As illustrated in Figure 1, the proposed SHLSA method comprises three core components: **(a) Hierarchical Feature Aggregation (HFA)** for robust feature extraction of *feature extractor*, **(b) Confidence-Aware Complementary Learning (CACL)** strategy with positive and negative pseudo-labels for high-quality pseudo-labels of *classifier*, and **(c) Stepwise Domain Alignment (SDA)** via entropy-based partitioning. We first train the **HFA** module on the target domain $D_t$ using local-global joint pseudo labels generated by HFA-EMA. Specially, by fusing features across hierarchical, HFA yields robust high-level representations that enhance object coverage and semantic consistency. Based on these features, **CACL** separates pseudo labels into positive and negative sets via softmax scores, enabling complementary learning from both confident and uncertain predictions.

During training, we compute the self-entropy of each sample and maintain an Entropy Bank via momentum-based updates. These entropy estimates are then used by **SDA** to partition the target domain into a pseudo-source subset $D_{ps}$ and a residual subset $D_{rt}$. To align the source and target domains, SDA first aligns pseudo-source features (HFA-PS-S) extracted by the task-specific HFA* with general semantics (PRE-S) from a frozen ImageNet-pretrained PRE-NET*. This step refines the high-confidence pseudo-source features to better reflect transferable semantics. The corrected HFA-PS-S then serves as a semantic bridge to guide low-confidence target features (HFA-RT-S) via a MixMatch-based semi-supervised strategy. This stepwise alignment enables phased adaptation from reliable to ambiguous regions, enhancing high-level semantic consistency under domain shift.

### 3.3 Hierarchical feature aggregation

High-level semantic alignment is crucial for TTA, especially under large domain shifts with diverse object appearance and layout. Non-hierarchical representations, whether global or local, often miss important cues, hindering dense prediction. We introduce a **hierarchical feature aggregation module** that fuses local and global information across abstraction levels. A shared extractor encodes complementary spatial and semantic cues, producing both fine-grained and holistic representations.

Formally, let $\boldsymbol{x} \in \mathcal{X}_t$ denote a target-domain input sample. To capture global semantic context, we downsample $\boldsymbol{x}$ and compute a coarse-grained prediction $P_{\text{global}}$. To extract fine-grained local semantics, $\boldsymbol{x}$ is divided into overlapping local regions indexed by $\text{Grid}(\boldsymbol{x})$ in the spatial feature space, with each region processed by a shared backbone to extract localized features:

$$f_{\text{local}_i} = \{\mathbf{f}\left(\boldsymbol{x}_{r_i}\right)\}, \quad r_i \in \text{Grid}(\boldsymbol{x}), \tag{2}$$

Each patch-level prediction $P_i = \mathbf{g}(f_{\text{local}_i})$ is independently decoded using a forzen classifier $\mathbf{g}$, based on features extracted by shared encoder $\mathbf{f}$. These predictions are then aggregated to reconstruct a detailed, locally-aware output $P_{\text{local}}$. To properly handle overlapping regions, we maintain a count matrix that records the number of times each pixel is covered by a patch. The aggregated prediction is normalized by this count matrix Mask to ensure consistent pixel-wise contributions:

$$P_{\text{local}} = \frac{\sum_i \text{Pad}(P_i)}{\sum_i \text{Mask}_i}. \tag{3}$$

To effectively integrate both global and local semantic cues, we introduce a **semantic-level attention mechanism** to adaptively fuse $P_{\text{global}}$ and $P_{\text{local}}$. The fusion is defined as:

$$P_{\text{fused}} = \boldsymbol{A} \cdot P_{\text{local}} + (\mathbf{1} - \boldsymbol{A}) \cdot \text{Align}(P_{\text{global}}). \tag{4}$$

Here, $\boldsymbol{A}_{ij} \in [0, 1]$ denotes the attention weight that adaptively balances the contributions between global and local predictions. The operator $\text{Align}(\cdot)$ harmonizes their feature representations, where $P_{\text{global}}$ and $P_{\text{local}}$ jointly form the final prediction through this learned attention mechanism. For more detailed implementation details, please refer to Appendix A.4.1.

This framework fuses global context with fine-grained details, reinforcing semantic consistency and enabling robust cross-domain feature alignment for diverse visual tasks.

### 3.4 Confidence-aware complementary learning strategy

To enhance high-level semantic alignment between pseudo-source and target domains, we propose a **confidence-aware complementary learning** strategy. Based on hierarchical pseudo-labels, we identify *positive* classes with high confidence and *negative* classes confidently rejected, providing complementary supervision that captures richer semantics and suppresses noisy predictions.

In this learning strategy, let $C$ be the number of classes, and let $\boldsymbol{p} = [p_1, p_2, \ldots, p_C] \in \mathbb{R}^C$ denote the predicted class probability distribution for a given input sample, satisfying:

$$\sum_{i=1}^{C} p_i = 1, \quad p_i \geq 0. \tag{5}$$

A confidence threshold $\tau_{\text{pos}}$ is introduced to identify positive predictions, where class $i$ is considered positive if $p_i \geq \tau_{\text{pos}}$. To justify the separation of predictions into positive and negative subsets, we introduce an entropy-based analysis in **Theorem** 1, whose proof is detailed in Appendix A.2.

Building on this, the theorem demonstrates that low-entropy predictions allow confident partitioning of class probabilities into positive and negative subsets, providing a foundation for selective learning from high-confidence predictions. Beyond fixed thresholds, we propose an adaptive strategy that leverages the relative structure of $\boldsymbol{p}$. Specially, high-confidence samples with a sharp decay in class probabilities are straightforward to classify, while low-confidence samples with flatter distributions introduce more ambiguity. By exploiting these differences, the confidence-aware strategy leverages these differences to assign more negative labels to high-confidence samples and fewer to low-confidence ones, enabling robust learning from both confident and uncertain predictions.

**Theorem 1** (Confidence-Separated Hypothesis Support Bound). *Let $\boldsymbol{p} \in \Delta^{C-1}$ be a categorical distribution over label space $\mathcal{Y} = \{1, \ldots, C\}$, with entropy bounded by $\mathcal{H}(\boldsymbol{p}) \leq H_0$. Then for any $\alpha \in (0, 1)$, there exist thresholds $\tau_\alpha > \tau_\beta$ such that:*

$$\mathcal{Y}_+ := \{c \mid p_c \geq \tau_\alpha\}, \quad \mathcal{Y}_- := \{c \mid p_c \leq \tau_\beta\}, \quad \mathcal{Y}_0 := \mathcal{Y} \setminus (\mathcal{Y}_+ \cup \mathcal{Y}_-), \tag{6}$$

*satisfying:*

$$\mathbb{E}_{c \sim \boldsymbol{p}}[\log p_c \mid c \in \mathcal{Y}_+] - \mathbb{E}_{c \sim \boldsymbol{p}}[\log(1 - p_c) \mid c \in \mathcal{Y}_-] \geq \kappa(H_0, \tau_\alpha), \quad (7)$$

$$\sum_{c \in \mathcal{Y}_-} p_c \leq \epsilon(H_0), \quad where \ \epsilon(H_0) \to 0 \ as \ H_0 \to 0. \quad (8)$$

To capture the relative confidence among class predictions, we first sort the predicted probabilities in descending order:

$$\boldsymbol{p}_{\text{sorted}} = [p_{(1)}, p_{(2)}, \ldots, p_{(C)}], \quad p_{(1)} \geq p_{(2)} \geq \cdots \geq p_{(C)}. \quad (9)$$

Based on the identified drop, we construct a ternary mask $\mathbf{m} \in \{-1, 0, 1\}^C$ to categorize each class $j$, and use it to define the *complementary learning loss*:

$$m_j = \begin{cases} 1, & \text{if } p_j \geq \tau_{\text{pos}}, \\ -1, & \text{if } j > i^*, \quad \text{where } i^* = \min\{i | r_i \geq \tau_{\text{neg}}\}, \\ 0, & \text{otherwise}, \end{cases} \quad (10)$$

$$\mathcal{L}_{CACL} = -\frac{1}{n_t} \sum_{i=1}^{n_t} \sum_{j=1}^{|\boldsymbol{x}_{t,i}|} \sum_{c=1}^{C} \left[ \mathbf{1}_{(\text{m}_j=1)} \log p_{\boldsymbol{x}_t}^{(i,j,c)} + \mathbf{1}_{(\text{m}_j=-1)} \log(1 - p_{\boldsymbol{x}_t}^{(i,j,c)}) \right], \quad (11)$$

where $\mathbf{1}_{(\cdot)}$ denotes the indicator function, and $|\boldsymbol{x}_{t,i}|$ represents the number of prediction units (e.g., instances or positions) in the $i$-th target sample.

By leveraging relative confidence gaps to distinguish confident predictions from rejections, this strategy improves semantic discrimination and enhances learning under domain shift. **Notably**, rather than treating unselected classes as positives, we apply an absolute threshold $\tau_{\text{pos}}$ to select confident positives. This approach avoids misclassifying ambiguous, hard-to-define classes as positives, ensuring reliable learning from clear, high-confidence labels.

### 3.5 STEPWISE DOMAIN ALIGNMENT

During HFA module updates, the uncertainty of each target sample is measured by **self-entropy**. As the model is updated using Exponential Moving Average (EMA), self-entropy captures the distributional divergence between the target and source domains, with low-entropy samples being more similar to the source domain and high-entropy samples reflecting larger domain shifts.

To enable this partitioning, we maintain an *Entropy Bank* that tracks the self-entropy of each sample throughout training. Similar to EMA, it updates the entropy value $\mathcal{H}_t(\boldsymbol{x})$ at each iteration $t$ by:

$$\mathcal{H}_t(\boldsymbol{x}) = \alpha \cdot \mathcal{H}_{t-1}(\boldsymbol{x}) + (1 - \alpha) \cdot H(p(\boldsymbol{x})), \quad (12)$$

where $H(p(\boldsymbol{x}))$ represents the current entropy computed from the predicted distribution $p(\boldsymbol{x})$, and $\alpha \in [0, 1)$ is the momentum coefficient. This EMA-based update smooths entropy fluctuations, providing a stable estimate of sample uncertainty. After computing self-entropy, the former $\tau_{\text{par}}$ is assigned as the *pseudo-source domain*, with the remaining samples as the *reamining target domain*.

After constructing the pseudo-source domain, we employ a **stepwise semantic alignment** strategy. The *pseudo-source domain $D_{ps}$* is aligned with general semantics from an ImageNet-pretrained model, refining high-confidence pseudo-source features. These corrected features then guide the adaptation of low-confidence *remaining target domain $D_{ps}$* samples, facilitating a smooth transition from reliable to ambiguous regions and enhancing semantic consistency under domain shift.

**Semantic feature alignment via pretrained supervision.** To ensure semantic consistency and structural alignment between the $D_s$ and the $D_{ps}$, we introduce a feature alignment regularization that leverages universal semantic features extracted from a pretrained visual backbone. This approach guides the pseudo-source domain toward the semantic space of the source domain, promoting more coherent class alignment even in the presence of domain shifts.

Given an input sample $\boldsymbol{x}$, let $f(\boldsymbol{x}) \in \mathbb{R}^{C \times H \times W}$ represent the feature map generated by the current model, and $f^{\text{pre}}(\boldsymbol{x}) \in \mathbb{R}^{C \times H \times W}$ the corresponding feature map extracted from a frozen pretrained

model. To enforce alignment between the features of the model and the pretrained model, we define a general feature alignment loss as follows:

$$\mathcal{L}_{\text{dis}} = \lambda_{\text{align}} \cdot \frac{1}{|\Omega|} \sum_{i \in \Omega} \text{Dist}\left(f(\boldsymbol{x})_i, f^{\text{pre}}(\boldsymbol{x})_i\right), \tag{13}$$

where $\text{Dist}(\cdot, \cdot)$ denotes the cosine similarity between normalized feature vectors, used to measure feature-level affinity. The set $\Omega \subseteq \{1, \dots, H \times W\}$ denotes the indices of spatial locations considered for feature alignment, selected based on class-specific or task-specific criteria:

$$\Omega = \{i \mid y_i \in \mathcal{C}_{\text{align}}\}, \tag{14}$$

where $y_i$ represents the available label or pseudo-label for location $i$, and $\mathcal{C}_{\text{align}}$ specifies the set of classes or regions targeted for feature alignment.

This loss encourages the model to match its feature representations with those from the pretrained model, guiding the pseudo-source domain toward the source semantic structure and enhancing cross-domain feature consistency. For example, even if the model initially misclassifies a *bus* as a *truck*, alignment with general semantic features (e.g., *vehicle*) extracted from the pretrained model helps steer the representation toward the correct class. This guidance improves the reliability of the pseudo-source domain and lays a more stable foundation for subsequent adaptation.

**Semantic alignment via mixed pseudo supervision.** To facilitate semantic transfer from the $D_{ps}$ to the $D_{rt}$, we employ a class-aware feature mixing strategy within a semi-supervised learning framework. Specifically, we utilize pseudo-labels $y_{ps}$ and $y_{rt}$ generated for high- and low-confidence samples, respectively, to construct a class mask $\boldsymbol{M}$ that defines region-wise semantic dominance. This mask enables interpolation between pseudo-source and target features at both the input and label levels. The resulting mixed sample $(\tilde{\boldsymbol{x}}_{\text{mix}}, \tilde{y}_{\text{mix}})$ is defined as:

$$\tilde{\boldsymbol{x}}_{\text{mix}} = \boldsymbol{M} \cdot \boldsymbol{x}_{ps} + (\boldsymbol{1} - \boldsymbol{M}) \cdot \boldsymbol{x}_{rt}, \quad \tilde{y}_{\text{mix}} = \boldsymbol{M} \cdot y_{ps} + (\boldsymbol{1} - \boldsymbol{M}) \cdot y_{rt}, \tag{15}$$

where $\boldsymbol{x}_{\text{ps}}$ and $\boldsymbol{x}_{\text{rt}}$ denote feature inputs from pseudo-source and target samples, and $y_{\text{ps}}, y_{\text{rt}}$ are their respective pseudo-labels. By enforcing prediction consistency on these mixed samples, the model is guided to propagate semantic structure from more reliable pseudo-source regions into uncertain target regions, improving decision boundary refinement and domain generalization.

By leveraging the pseudo-source domain, which encapsulates high-confidence, semantically reliable regions, the mixed samples are infused with informative supervision that guides the learning of uncertain target representations. These mixed inputs are then used to optimize the model via a cross-entropy loss using CACL:

$$\mathcal{L}_{\text{mix}} = \text{CACL}(-\sum_c \tilde{y}_{\text{mix}}^{(c)} \log p(c \mid \tilde{\boldsymbol{x}}_{\text{mix}})), \tag{16}$$

where $c$ indexes the semantic classes and $p(c \mid \tilde{\boldsymbol{x}}_{\text{mix}})$ denotes the predicted class probability from the model. This objective encourages consistency between the model's predictions and the interpolated supervision, refining decision boundaries under domain shift.

Through this semi-supervised feature interpolation, the model progressively aligns uncertain target features with the more structured semantics of the pseudo-source. This process narrows the distributional gap and enhances generalization by leveraging pseudo-source guidance for the adaptation of less confident regions in the target domain.

## 4 EXPERIMENTS

### 4.1 DATASETS AND EVALUATION MTRICS

We evaluate SHLSA on semantic segmentation and image classification benchmarks. Segmentation, with its dense pixel-level supervision and higher cross-domain difficulty, provides a rigorous testbed. We use GTA5→Cityscapes, SYNTHIA→Cityscapes, and Cityscapes→ACDC: GTA5 and SYNTHIA are synthetic datasets with pixel-level labels, while Cityscapes and ACDC contain real urban scenes, the latter emphasizing challenging conditions (e.g., night, fog). Performance is measured by mean IoU. For classification, we adopt VisDA-C, Office-Home, and Office-31, covering datasets from small- to large-scale with diverse domain shifts, and report Top-1 accuracy. For more comparison methods and dataset information, please refer to Appendix A.4.2.

Table 1: Semantic segmentation performance of SHLSA on different tasks.

| Method | SF | road | side. | build. | wall | fence | pole | light | sign | vege. | terr. | sky | person | rider | car | truck | bus | train | motor. | bike | mIoU |
|---|---|---|---|---|---|---|---|---|---|---|---|---|---|---|---|---|---|---|---|---|---|
| **GTA5 → Cityscapes (CS)** | | | | | | | | | | | | | | | | | | | | | |
| TransDA-B | ✗ | 94.7 | 64.2 | 89.2 | 48.1 | 45.8 | 50.1 | 60.2 | 40.8 | 90.4 | 50.2 | 93.7 | 76.7 | 47.6 | 92.5 | 56.8 | 60.1 | 47.6 | 49.6 | 55.4 | 63.9 |
| DAFormer | ✗ | 95.7 | 70.2 | 89.4 | 53.5 | 48.1 | 49.6 | 55.8 | 59.4 | 89.9 | 47.9 | 92.5 | 72.2 | 44.7 | 92.3 | 74.5 | 78.2 | 65.1 | 55.9 | 61.8 | 68.3 |
| HRDA | ✗ | 96.4 | 74.4 | 91.0 | 61.6 | 51.5 | 57.1 | 63.9 | 69.3 | 91.3 | 48.4 | 94.2 | 79.0 | 52.9 | 93.9 | 84.1 | 85.7 | 75.9 | 63.9 | 67.5 | 73.8 |
| IDM | ✗ | 97.2 | 77.1 | 89.8 | 51.7 | 51.7 | 54.5 | 59.7 | 64.7 | 89.2 | 45.3 | 90.5 | 74.2 | 46.6 | 92.3 | 76.9 | 59.6 | 81.2 | 57.3 | 62.4 | 69.5 |
| DAFormer | ✓ | 87.7 | 33.4 | 83.9 | 28.1 | 27.5 | 35.9 | 42.9 | 28.7 | 82.4 | 28.6 | 83.1 | 65.0 | 37.0 | 85.8 | 53.9 | 46.3 | 31.8 | 23.6 | 36.8 | 49.6 |
| HRDA | ✓ | 83.3 | 28.2 | 83.3 | 43.3 | 22.2 | 42.9 | 47.7 | 38.2 | 87.2 | 40.0 | 81.6 | 69.5 | 35.9 | 84.8 | 42.7 | 50.4 | 41.2 | 33.7 | 29.6 | 51.9 |
| IDM | ✓ | 93.9 | 59.1 | 86.6 | 35.3 | 30.4 | 42.2 | 45.1 | 57.8 | 88.4 | 35.1 | 89.4 | 69.7 | 39.8 | 89.1 | 66.8 | 46.0 | 13.5 | 41.1 | 61.2 | 57.4 |
| ATP | ✓ | **96.6** | **75.3** | **89.4** | 50.2 | **41.5** | 47.5 | 48.6 | 61.1 | 89.8 | **48.3** | **93.4** | 70.4 | 40.1 | 89.8 | 66.7 | 58.2 | 30.3 | 53.4 | **65.6** | 64.0 |
| SHLSA | ✓ | 93.0 | 61.0 | 88.8 | **51.8** | 33.9 | **54.3** | **62.5** | **66.5** | 89.8 | 43.3 | 92.5 | **78.1** | **47.2** | **93.0** | **78.6** | **79.6** | **74.6** | **63.8** | 63.4 | **69.2** |
| **SYNTHIA → Cityscapes (CS)** | | | | | | | | | | | | | | | | | | | | | |
| TransDA-B | ✗ | 90.4 | 54.8 | 86.4 | 31.1 | 1.7 | 53.8 | 61.1 | 37.1 | 90.3 | - | 93.0 | 71.2 | 25.3 | 92.3 | - | 66.0 | - | 44.4 | 49.8 | 59.3 |
| DAFormer | ✗ | 84.5 | 40.7 | 88.4 | 41.5 | 6.5 | 50.0 | 55.0 | 54.6 | 86.0 | - | 89.8 | 73.2 | 48.2 | 87.2 | - | 53.2 | - | 53.9 | 61.7 | 60.9 |
| HRDA | ✗ | 85.2 | 47.7 | 88.8 | 49.5 | 4.8 | 57.2 | 65.7 | 60.9 | 85.3 | - | 92.9 | 79.4 | 52.8 | 89.0 | - | 64.7 | - | 63.9 | 64.9 | 65.8 |
| DAFormer | ✓ | 64.3 | 25.1 | 78.5 | 23.8 | **1.9** | 37.3 | 29.7 | 22.8 | 80.4 | - | 83.0 | 65.1 | 26.6 | 69.8 | - | 38.3 | - | 22.7 | 32.8 | 43.8 |
| HRDA | ✓ | 72.2 | 26.6 | 80.8 | 23.0 | 0.5 | 42.5 | 41.0 | 31.5 | 84.3 | - | 86.2 | 64.3 | 29.3 | 73.5 | - | 28.8 | - | 12.4 | 41.6 | 46.1 |
| IDM | ✓ | 82.2 | 37.9 | 83.5 | 20.3 | 1.5 | 47.3 | 41.7 | 25.6 | 84.4 | - | 86.8 | 61.6 | 25.0 | 87.6 | - | 43.7 | - | 30.2 | 36.4 | 49.7 |
| MISFIT | ✓ | 80.2 | 38.5 | 85.9 | 30.3 | 1.2 | 52.3 | 56.8 | 29.0 | **89.9** | - | 88.3 | 68.1 | 10.8 | **92.1** | - | **69.0** | - | 26.3 | 52.6 | 54.5 |
| ATP | ✓ | **90.6** | **54.4** | 86.7 | 28.5 | 0.5 | 50.3 | 52.4 | 50.5 | 87.4 | - | **93.4** | 70.2 | 35.8 | 89.6 | - | 53.5 | - | 50.6 | 51.1 | 59.1 |
| SHLSA | ✓ | 90.0 | 51.9 | **87.0** | **35.4** | 1.0 | **56.7** | **64.6** | 53.6 | 88.5 | - | 92.3 | **78.3** | **44.4** | 89.8 | - | 63.4 | - | **62.8** | **65.1** | **64.1** |
| **Cityscapes (CS) → ACDC** | | | | | | | | | | | | | | | | | | | | | |
| TENT | ✓ | 85.3 | 50.2 | 85.4 | 45.4 | 32.7 | 50.4 | 59.4 | 66.1 | 86.4 | 45.7 | 97.5 | 57.9 | 53.8 | 84.7 | 51.0 | 66.9 | 72.4 | 40.2 | 50.1 | 62.2 |
| CoTTA | ✓ | 85.7 | 50.9 | **85.9** | 45.9 | 33.6 | 54.8 | 62.3 | **69.9** | 87.1 | 45.7 | **97.7** | 63.3 | **59.4** | 85.1 | 52.8 | 68.0 | 74.1 | 44.9 | 55.1 | 64.4 |
| DePT | ✓ | 85.0 | 50.6 | 85.5 | 45.7 | 33.2 | 53.9 | 61.6 | 69.4 | 86.7 | 45.5 | 97.4 | 62.6 | 59.2 | 85.1 | 52.5 | 68.0 | 73.7 | 44.3 | 54.3 | 63.9 |
| VDP | ✓ | 85.7 | 50.9 | **85.9** | 45.9 | 33.6 | 54.8 | 62.2 | **69.9** | 87.0 | 45.7 | 97.6 | 63.3 | 59.2 | 85.1 | 52.8 | 68.1 | 74.1 | 44.8 | **54.9** | 64.3 |
| IDM | ✓ | **88.8** | **63.2** | 85.8 | 45.5 | 30.3 | 42.1 | 69.7 | 62.7 | **87.4** | **51.7** | 96.5 | 61.8 | 29.3 | 86.5 | 80.5 | 68.9 | 70.1 | **57.0** | 52.8 | 64.8 |
| SHLSA | ✓ | 84.1 | 54.7 | 84.6 | **51.7** | **43.7** | **61.5** | **72.5** | 59.5 | 72.2 | 38.5 | 79.1 | **63.8** | 44.6 | **88.0** | **84.4** | **76.6** | **78.3** | 48.2 | 53.4 | **65.2** |

## 4.2 IMPLEMENTATION DETAILS

For semantic segmentation, we use MiT-B5 Xie et al. (2021) as the backbone, and ResNet-50/101 for classification. Segmentation is implemented in MMSegmentation Contributors (2020), while classification follows the SHOT Liang et al. (2020) codebase. Optimization employs SGD with learning rate $2.5 \times 10^{-4}$, momentum 0.9, and weight decay $5 \times 10^{-4}$, using a *poly* schedule $(1 - \frac{iter}{max\_iter})^{0.9}$. During entropy minimization the classifier is frozen; otherwise it is updated with learning rate $2.5 \times 10^{-3}$. Hyperparameters are chosen via an unsupervised strategy (see Section 4.5). **Note**: Test-Time Domain Adaptation (TTDA), also known as Source-Free Domain Adaptation (SFDA) Liang et al. (2025), is discussed in TTA experiments, with online TTA results referenced in the Appendix A.4.7. For more details, please refer to Appendix A.4.1.

## 4.3 RESULTS OF SEMANTIC SEGMENTATION

We evaluate SHLSA on three UDA benchmarks: GTA5→Cityscapes, SYNTHIA→Cityscapes, and Cityscapes→ACDC (Table 1), to verify its ability to reduce **class confusion** and stabilize **decision boundaries** beyond low-level SDE alignment. SHLSA achieves state-of-the-art results in all settings under the source-free constraint.

**GTA5→Cityscapes:** SHLSA attains 69.2 mIoU, +5.2 over the prior best, with gains in frequent classes (*car*, *road*) and fine-grained or ambiguous ones (*person* 78.1, *bus* 79.6, *train* 74.6). Texture-sensitive classes (*wall*, *light*, *sign*) also improve, showing more **stable boundaries** than low-level cues alone. **SYNTHIA→Cityscapes:** Despite larger content/layout gaps, SHLSA delivers 64.1 mIoU (16-class), surpassing all source-free methods. Better scores on *pole* (56.7), *rider* (44.4), and *motorcycle* (62.8) indicate that **semantic alignment alleviates boundary instability** typical of shallow features. **Cityscapes→ACDC:** Under fog, night, and rain, SHLSA reaches 65.2 mIoU, preserving **semantic separability** even when low-level cues degrade.

Overall, SHLSA's semantic-aware stepwise alignment reduces **class confusion** and **unstable boundaries** beyond SDE, enabling reliable performance across diverse urban scenes. Visualization and qualitative analyses appear in Appendix A.4.3.

## 4.4 RESULTS OF IMAGE CLASSIFICATION

We evaluate SHLSA on Office-31, Office-Home and VisDA-C under source-free settings (Table 2) to assess its ability to reduce **class confusion** and enhance **decision stability** beyond low-level feature alignment. In addition, we have also conducted tests on TTA for multi-source domains (MSDA) and multi-target domains (MTDA), the results are presented in the Table 3. On **Office-31** it reaches

Table 2: Image classification performance of SHLSA on different tasks.

| Office-31 (ResNet50 backbone) | | | | | | | | |
|---|---|---|---|---|---|---|---|---|
| Method | SF | A→D | A→W | D→A | D→W | W→A | W→D | Avg |
| SHOT | ✓ | 93.7 | 91.1 | 74.2 | 98.2 | 74.6 | **100.** | 88.6 |
| DIFO | ✓ | **97.2** | 95.5 | 83.0 | 97.2 | 83.2 | 98.8 | 92.5 |
| ProDe | ✓ | 96.8 | 96.4 | **83.1** | 97.0 | **82.5** | 99.8 | 92.6 |
| SHLSA | ✓ | 97.0 | **96.9** | 82.6 | **98.2** | 82.1 | 99.8 | **92.8** |

| Office-Home (ResNet50 backbone) | | | | | | | | | | | | | |
|---|---|---|---|---|---|---|---|---|---|---|---|---|---|
| Method | SF | Ar→Cl | Ar→Pr | Ar→Rw | Cl→Ar | Cl→Pr | Cl→Rw | Pr→Ar | Pr→Cl | Pr→Rw | Rw→Ar | Rw→Cl | Rw→Pr | Avg. |
| PDA | ✗ | 55.4 | 85.1 | 85.8 | 75.2 | 85.2 | 85.2 | 74.2 | 55.2 | 85.8 | 74.7 | 55.8 | 86.3 | 75.3 |
| DAMP | ✗ | 59.7 | 88.5 | 86.8 | 76.6 | 88.9 | 87.0 | 76.3 | 59.6 | 87.1 | 77.0 | 61.0 | 89.9 | 78.2 |
| SHOT | ✓ | 57.1 | 78.1 | 81.5 | 68.0 | 78.2 | 78.1 | 67.4 | 54.9 | 82.2 | 73.3 | 58.8 | 84.3 | 71.8 |
| DIFO | ✓ | 70.6 | 90.6 | 88.8 | 82.5 | 90.6 | 88.8 | 80.9 | 70.1 | 88.9 | 83.4 | 70.5 | 91.2 | 83.1 |
| ProDe | ✓ | 72.7 | **92.3** | 90.5 | 82.5 | 91.5 | 90.7 | 82.5 | 72.5 | **90.8** | 83.0 | 72.6 | 92.2 | 84.5 |
| SHLSA | ✓ | **73.1** | 91.9 | **91.2** | **84.0** | **91.6** | **90.8** | **82.8** | **73.7** | 90.7 | **83.4** | **74.7** | **92.4** | **85.0** |

| VisDA-C (ResNet101 backbone) | | | | | | | | | | | | | |
|---|---|---|---|---|---|---|---|---|---|---|---|---|---|
| Method | SF | plane | bike | bus | car | horse | knife | mcycle | person | plant | sktbrd | train | truck | Avg. |
| STAR | ✗ | 95.0 | 84.0 | 84.6 | 73.0 | 91.6 | 91.8 | 85.9 | 78.4 | 94.4 | 84.7 | 87.0 | 42.2 | 82.7 |
| RWOT | ✗ | 95.1 | 80.3 | 83.7 | 90.0 | 92.4 | 68.0 | 92.5 | 82.2 | 87.9 | 78.4 | 90.4 | 68.2 | 84.0 |
| SHOT | ✓ | 94.3 | 88.5 | 80.1 | 57.3 | 93.1 | 94.9 | 80.7 | 80.3 | 91.5 | 89.1 | 86.3 | 58.2 | 82.9 |
| PLUE | ✓ | 97.3 | 96.2 | 90.5 | **91.8** | 97.0 | 87.4 | 87.7 | 97.0 | 84.3 | 93.0 | 81.0 | 90.0 |
| ATP | ✓ | 97.6 | 91.8 | **88.7** | 73.1 | 97.6 | 92.9 | 92.0 | **95.7** | 93.4 | 89.0 | 87.9 | 71.3 | 89.3 |
| ProDe | ✓ | 98.3 | **92.4** | 86.6 | 80.5 | 98.1 | **98.0** | 92.3 | 84.3 | 94.7 | **97.0** | 94.1 | **75.6** | 91.0 |
| **SHLSA** | ✓ | **98.3** | 90.0 | 88.0 | 87.2 | **98.3** | 97.3 | **93.5** | 86.3 | **97.7** | 96.6 | **95.7** | 66.8 | **91.3** |

92.8% across 6 shifts. On **Office-Home**, SHLSA attains 85.0% average accuracy over 12 domain shifts, while on **VisDA-C**, SHLSA achieves 91.3% average accuracy, outperforming prior methods through multi-level semantic alignment. Strong results on *plane*, *horse*, and *train* show how hierarchical feature aggregation improves semantic separability. Combined with the results shown in the Figure 2 These results confirm that our **CACL** and **HFA** modules jointly strengthen semantic alignment and yield **more reliable decision boundaries**. Moreover, under the TTA Settings of multiple-source domains and multiple-target domains, SHLSA has shown SOTA performance. Additional qualitative analyses and **t-SNE visualizations** appear in Appendices A.4.4 and A.4.5.

Table 3: Multi-Source and Multi-Target performance of SHLSA on Office-Home.

| Task | Method | Ar | Cl | Pr | Rw | Avg. |
|---|---|---|---|---|---|---|
| **MTDA** | CoNMix | 75.6 | 81.4 | 71.4 | 73.4 | 75.4 |
| | ProDe | 83.3 | 89.2 | 80.9 | 81.2 | 83.6 |
| | **SHLSA** | **84.1** | **89.4** | **81.6** | **81.6** | **84.2** |
| **MSDA** | SHOT-Ens | 82.9 | 82.8 | 59.3 | 72.2 | 74.3 |
| | DECISION | 83.6 | 84.4 | 59.4 | 74.5 | 75.5 |
| | ProDe | 91.1 | 92.5 | 73.4 | 83.0 | 85.0 |
| | **SHLSA** | **91.7** | **93.1** | **75.2** | **83.6** | **85.9** |

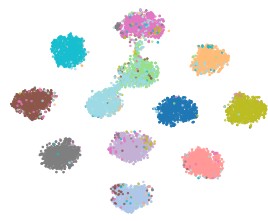

Figure 2: SHLSA t-SNE visualization.

## 4.5 SENSITIVITY ANALYSIS

We analyze the sensitivity of three hyperparameters on VisDA-C: the negative threshold $\tau_{neg}$, positive threshold $\tau_{pos}$, and partition threshold $\tau_{par}$, which control negative selection, confident positives, and data partitioning. As shown in Figure 3, performance is most sensitive to $\tau_{neg}$, with values below 0.5 causing sharp drops due to noisy negatives. $\tau_{pos}$ is comparatively stable, while $\tau_{par}$ shows steady gains, peaking at 0.8–0.9. We thus set $\tau_{neg} = 0.9$, $\tau_{pos} = 0.9$, and $\tau_{par} = 0.8$ in all experiments. Appendix A.3 further confirms these thresholds can be tuned independently, and Appendix A.5 provides more detailed discussion with mitigation plans.

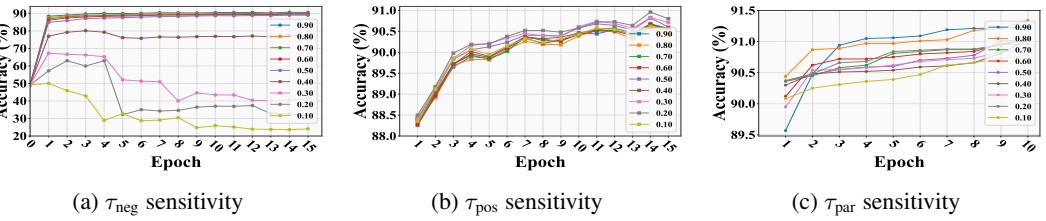

(a) $\tau_{neg}$ sensitivity      (b) $\tau_{pos}$ sensitivity      (c) $\tau_{par}$ sensitivity

Figure 3: Sensitivity analysis for the thresholds.

## 4.6 ABLATION STUDY

As shown in Table 4, introducing **HFA** alone boosts segmentation performance from 44.5 to 57.5 mIoU by aggregating local and global features, which enhances high-level semantic representations; however, it shows minimal impact on classification (84.6%), suggesting spatial context is more crucial in dense prediction tasks (more analysis is in the Appendix A.7). Adding **CACL** further improves both segmentation (65.6 mIoU) and classification (84.7%) by leveraging high-confidence predictions and uncertain regions to provide complementary supervision. Finally, **SDA** brings consistent gains across tasks, with segmentation mIoU reaching 69.2 and classification accuracy improving to 85.0%, highlighting the effectiveness of

Table 4: Ablation studies on segmentation (above) and classification (below).

(a) Semantic segmentation on GTA5→Cityscapes

| Method | HFA | CACL | SDA | mIoU |
|---|---|---|---|---|
| Baseline | | | | 44.5 |
| (a) | ✓ | | | 57.5 |
| (b) | ✓ | ✓ | | 65.6 |
| (c) | ✓ | ✓ | ✓ | 69.2 |

(b) Image classification on Office-Home

| Method | HFA | CACL | SDA | Acc |
|---|---|---|---|---|
| Baseline | | | | 84.5 |
| (a) | ✓ | | | 84.6 |
| (b) | ✓ | ✓ | | 84.7 |
| (c) | ✓ | ✓ | ✓ | 85.0 |

entropy-aware domain partitioning and progressive alignment. These results collectively demonstrate the complementary strengths of the three modules and the versatility of SHLSA across heterogeneous tasks. More detailed results are in the Appendix A.4.6.

## 4.7 DATA LEAKAGE AND COMPUTATIONAL COST ANALYSIS

To verify that using a pre-trained model to correct the *pseudo-source domain* in **SDA** does not introduce extra domain priors, we test different target domains with both the pre-trained and source models. **Note that** the pre-trained model keeps its feature extractor, replacing only the classifier with that of the source model. As shown in Table 5, the source model consistently outperforms the ImageNet-pre-trained model across all source–target pairs, indicating that the pre-trained model provides generic semantic knowledge (e.g., *car* for *van*) without introducing domain-specific information or acting as a teacher, thus it will not cause data leakage.

Table 5: Performance of Source (outside) and Pre-Trained (inside) Models.

| Source Domain | Target Domain | | | | Avg |
|---|---|---|---|---|---|
| | Ar | Cl | Pr | Rw | |
| Ar | 97.50 (90.32) | 44.80 (33.17) | 65.88 (63.62) | 72.11 (71.72) | **70.07** (64.71) |
| Cl | 46.86 (54.26) | 97.06 (57.39) | 58.69 (59.04) | 61.07 (64.52) | **65.92** (58.80) |
| Pr | 49.91 (54.68) | 40.20 (32.46) | 99.27 (91.28) | 71.55 (71.38) | **65.23** (62.45) |
| Rw | 62.43 (64.28) | 46.44 (35.60) | 76.80 (71.57) | 98.02 (91.05) | **70.92** (65.63) |

In terms of computational cost, as a core component of SHLSA, although **HFA** greatly enhances the quality of semantic features in TTA scenarios, especially for semantically intensive tasks such as semantic segmentation, its introduction may increase the computational resources required. We provide both qualitative and quantitative analyses of this aspect in the Appendix A.6.

## 5 CONCLUSION

In this paper, we propose SHLSA, a stepwise feature alignment framework for test-time adaptation that focuses on high-level semantic alignment to address class confusion and unstable decision boundaries common in low-level methods. By combining hierarchical semantic aggregation with confidence-aware complementary learning, SHLSA enhances semantic consistency and robustness under domain shifts. The stepwise process uses universal semantic priors to progressively bridge source, pseudo-source, and target domains, effectively narrowing domain gaps. The approach's effectiveness is validated through extensive experiments on synthetic and real-world TTA datasets.

**Limitations.** Despite strong results, SHLSA has limitations. The fixed local-to-global context ratio may restrict adaptability across tasks with varying semantic granularity and is less effective for single-label classification where spatial context is less informative. Future work could explore adaptive aggregation or extend SHLSA to more complex multi-label scenarios. Additionally, the fixed self-entropy threshold $\tau_{par}$ for pseudo-source domain construction may not generalize well across domains, highlighting the need for more flexible, data-driven thresholding.

ETHICS STATEMENT

This work does not involve human subjects, sensitive data, or applications that could raise ethical concerns. It does not present issues related to privacy, security, discrimination, bias, fairness, or potential misuse. Hence, we believe there are no specific ethics-related considerations in this paper.

REPRODUCIBILITY STATEMENT

The proposed SHLSA method is fully reproducible. The source code is available at `https://anonymous.4open.science/r/SHLSA`, with detailed instructions and key code snippets for the core operations of SHLSA provided in the README.

LLMS USAGE STATEMENT

In this work, Large Language Models (LLMs) were used solely for language refinement and proof-reading of the manuscript. They were not involved in research ideation, experimental design, data analysis, or the generation of scientific content. The authors take full responsibility for the final content of this paper.

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

# A  APPENDIX

## A.1  PSEUDO-CODE OF THE PROPOSED METHOD

---

**Algorithm 1** Stepwise High-Level Semantic Alignment (SHLSA)

---

**Require:** Target dataset $D_t$, source model $\mathcal{M}_s = (\mathbf{f_s}, \mathbf{g_s})$
**Ensure:** Adapted model $\mathcal{M}_t$
    *// Stage1: Update HFA module*
1: **while** not converged **do**
2:    **for all** $\boldsymbol{x} \in D_t$ **do**
3:        *// Hierarchical Feature Aggregation & Entropy Estimation*
4:        Extract hierarchical semantic features $\{\mathbf{f}^{(l)}(\boldsymbol{x}), \mathbf{f}^{(g)}(\boldsymbol{x})\}$ from $\mathbf{f_s}$
5:        Fuse local-global features via attention: $f_{\text{hfa}}(x) \leftarrow \text{AttnFusion}(\{\mathbf{f}^{(l)}(\boldsymbol{x}), \mathbf{f}^{(g)}(\boldsymbol{x})\})$
6:        Generate pseudo-probabilities $p(\boldsymbol{x}) \leftarrow \text{softmax}(\mathbf{g_s}(f_{\text{hfa}}(\boldsymbol{x})))$
7:        Update entropy bank: $\mathcal{H}(\boldsymbol{x}) \leftarrow \alpha \cdot \mathcal{H}(\boldsymbol{x}) + (1 - \alpha) \cdot H(p(\boldsymbol{x}))$
8:        *// Confidence-Aware Complementary Learning*
9:        Build confidence mask $m$ from ranked logits (Eq. 10)
10:      Compute $\mathcal{L}_{\text{cacl}}$ from confident and uncertain regions (Eq. 11)
11:      Update model using $\mathcal{L}_{\text{cacl}}$
12:    **end for**
13: **end while**
    *// Stage2: Stepwise Domain Alignment*
14: Partition $D_t$ into $D_{ps}$ and $D_{rt}$ using entropy threshold $\tau_{\text{par}}$
15: **while** not converged **do**
16:    **for all** $(\boldsymbol{x}_{ps}, \boldsymbol{x}_{rt}) \in \mathcal{D}_{ps} \times \mathcal{D}_{rt}$ **do**
17:        *// STEP1: Semantic feature alignment via pretrained supervision*
18:        Extract features $f_{\text{ps}}(\boldsymbol{x}^{ps})$ via HFA; extract $f^{\text{pre}}(\boldsymbol{x}_{ps})$ via frozen pretrained net
19:        Compute semantic alignment loss $\mathcal{L}_{\text{dis}}$ (Eq. 13)
20:        *// STEP2: Semantic alignment via mixed pseudo supervision*
21:        Generate class-masked mixed sample $(\tilde{\boldsymbol{x}}, \tilde{y})$ (Eq. 15)
22:        Compute cross-entropy loss $\mathcal{L}_{\text{mix}}$ on mixed sample using CACL (Eq. 16)
23:        Update model using $\mathcal{L}_{\text{dis}} + \mathcal{L}_{\text{mix}}$
24:    **end for**
25: **end while**
26: **return** $\mathcal{M}_t$

---

## A.2  PROOF OF THEOREM

*Proof.* Let $\boldsymbol{p} = (p_1, \ldots, p_C) \in \Delta^{C-1}$ be a categorical distribution with entropy $\mathcal{H}(\boldsymbol{p}) = -\sum_{c=1}^{C} p_c \log p_c \leq H_0$. Fix any $\alpha \in (0, 1)$. We aim to construct thresholds $\tau_\alpha > \tau_\beta$ such that the sets

$$\mathcal{Y}_+ := \{c \mid p_c \geq \tau_\alpha\}, \quad \mathcal{Y}_- := \{c \mid p_c \leq \tau_\beta\}, \tag{17}$$

satisfy the stated bounds.

We begin by bounding the low-confidence tail. For any $\tau \in (0, 1)$, define $\mathcal{Y}_{<\tau} := \{c \mid p_c \leq \tau\}$. Then

$$\mathcal{H}(\boldsymbol{p}) \geq - \sum_{c \in \mathcal{Y}_{<\tau}} p_c \log p_c \geq - \log \tau \sum_{c \in \mathcal{Y}_{<\tau}} p_c, \tag{18}$$

which implies

$$\sum_{c \in \mathcal{Y}_{<\tau}} p_c \leq \frac{H_0}{-\log \tau}. \tag{19}$$

Setting $\tau_\beta := \tau$, this yields the desired upper bound on $\sum_{c \in \mathcal{Y}_-} p_c$ with $\epsilon(H_0) := H_0 / -\log \tau_\beta \to 0$ as $H_0 \to 0$.

Next, for the high-confidence region, choose $\tau_\alpha \in (0, 1)$ such that $\mathcal{Y}_+$ is nonempty. Then, for all $c \in \mathcal{Y}_+$, $p_c \geq \tau_\alpha$ implies $\log p_c \geq \log \tau_\alpha$, and for $c \in \mathcal{Y}_-$, $p_c \leq \tau_\beta$ implies $1 - p_c \geq 1 - \tau_\beta$, so

$\log(1 - p_c) \geq \log(1 - \tau_\beta)$. Thus,

$$\mathbb{E}_{c \sim \boldsymbol{p}}[\log p_c \mid c \in \mathcal{Y}_+] \geq \log \tau_\alpha, \tag{20}$$

$$\mathbb{E}_{c \sim \boldsymbol{p}}[\log(1 - p_c) \mid c \in \mathcal{Y}_-] \geq \log(1 - \tau_\beta), \tag{21}$$

and therefore

$$\mathbb{E}_{c \sim \boldsymbol{p}}[\log p_c \mid c \in \mathcal{Y}_+] - \mathbb{E}_{c \sim \boldsymbol{p}}[\log(1 - p_c) \mid c \in \mathcal{Y}_-] \geq \log \tau_\alpha - \log(1 - \tau_\beta) := \kappa(H_0, \tau_\alpha). \tag{22}$$

This concludes the proof. $\qquad\square$

### A.3 PROOF OF INDEPENDENCE FOR PARAMETER TUNING

*Proof.* Let $\boldsymbol{p} = (p_1, \ldots, p_C) \in \Delta^{C-1}$ be the predicted softmax vector for a sample with entropy $\mathcal{H}(\boldsymbol{p}) = -\sum_{c=1}^{C} p_c \log p_c$. Let $\boldsymbol{p}_{\text{sorted}} = [p_{(1)}, p_{(2)}, \ldots, p_{(C)}]$ denote the sorted version of $\boldsymbol{p}$ as same as Eq. 9.

The threshold $\tau_{\text{pos}}$ is used to identify confident predictions:

$$\mathcal{Y}_+ := \{c \mid p_c \geq \tau_{\text{pos}}\}. \tag{23}$$

The choice of $\tau_{\text{pos}}$ depends only on the absolute values of $\boldsymbol{p}$ and does not affect its ordering or entropy.

The threshold $\tau_{\text{neg}}$ is based on the relative drop between adjacent sorted entries:

$$r_i := \frac{p_{(i)} - p_{(i+1)}}{p_{(i)}}, \quad i^* := \min\{i \mid r_i \geq \tau_{\text{neg}}\}, \tag{24}$$

from which we define the low-confidence region:

$$\mathcal{Y}_- := \{c \mid c > i^*\}. \tag{25}$$

Note that $\tau_{\text{neg}}$ depends only on the relative differences in the ordered vector $\boldsymbol{p}_{\text{sorted}}$ and is independent of any fixed threshold such as $\tau_{\text{pos}}$.

Now consider the partition threshold $\tau_{\text{par}}$, which operates at the sample level. Let $\mathcal{X}_{\text{ps}} := \{x \mid \mathcal{H}(\boldsymbol{p}_x) \leq \tau_{\text{par}}\}$ denote the subset of target samples with low entropy. Since $\mathcal{H}(\boldsymbol{p})$ is a symmetric function of the distribution and depends only on the global shape of $\boldsymbol{p}$ rather than pointwise values or ordering, the partitioning induced by $\tau_{\text{par}}$ is independent of the selection mechanisms for class-level masks.

To formalize this independence, observe that the three thresholding operations in our framework correspond to distinct mappings:

$$\boldsymbol{p} \mapsto \mathcal{Y}_+, \quad \boldsymbol{p}_{\text{sorted}} \mapsto \mathcal{Y}_-, \quad \boldsymbol{p} \mapsto \mathcal{H}(\boldsymbol{p}). \tag{26}$$

Here, $\mathcal{Y}_+$ is determined by a fixed threshold on the unnormalized softmax scores ($\tau_{\text{pos}}$), $\mathcal{Y}_-$ is based on relative gaps in the sorted probability vector using $\tau_{\text{neg}}$, and $\mathcal{H}(\boldsymbol{p})$ aggregates the entire distribution through a permutation-invariant functional. Each of these quantities responds only to changes in its respective threshold.

Therefore, varying $\tau_{\text{pos}}$ affects only the set $\mathcal{Y}_+$, changing $\tau_{\text{neg}}$ influences only the structure of $\mathcal{Y}_-$, and adjusting $\tau_{\text{par}}$ modifies only the partitioning of samples into $\mathcal{X}_{\text{ps}}$ and $\mathcal{X}_{\text{rt}}$. These quantities are used in distinct components of the framework. Specifically, $\mathcal{Y}_+$ and $\mathcal{Y}_-$ are involved in confidence-aware complementary learning, while $\mathcal{X}_{\text{ps}}$ is used in the progressive domain alignment module. As a result, the thresholds $\tau_{\text{pos}}$, $\tau_{\text{neg}}$, and $\tau_{\text{par}}$ are functionally independent and can be tuned separately without mutual interference. $\qquad\square$

### A.4 SUPPLEMENTARY ANALYSIS OF THE EXPERIMENT

#### A.4.1 EXPERIMENT DETAILS

All experiments were conducted on a single NVIDIA GeForce RTX 3090. Some necessary experimental parameters have already been provided in Section 4.2, and here we mainly supplement the implementation details of the Method in Section 3.

For the $\text{Grid}(\boldsymbol{x})$ operation in Eq. 2, we divide the input along the $x$ and $y$ directions with a fixed step to obtain the grid $\{y_{i1}, y_{i2}, x_{i1}, x_{i2}\}$, where each $r_i$ corresponds to a patch in $\boldsymbol{x}$ defined by four coordinates.

For the $\text{Pad}(P_i)$ operation in Eq. 3, we apply zero padding. For the $\text{Mask}_i$ operation in the same equation, we compute it by counting the number of pixels covered by the $i$-th local patch.

For the $\boldsymbol{A}$ operation in Eq. 4, we obtain it through an attention module. In semantic segmentation tasks, this module outputs a pixel-level attention map to represent the importance of local semantics, while in image classification tasks, it directly averages local and global semantics. The $\text{Align}(\cdot)$ operation in the same equation is implemented using a simple bilinear interpolation-based *resize* to align the sizes of local and global features. The resulting $P_{\text{fused}}$ is directly used as the input to CACL for post-processing.

For $M$ in Eq. 15, we adopt the MixMatch operation to generate semantic-level mixed data, following the procedure described in Berthelot et al. (2019), which we keep consistent with.

### A.4.2 COMPETITORS AND EVALUATION DATASETS

We adopt SHOT Liang et al. (2020) as the baseline for semantic segmentation, which utilizes source hypothesis transfer for source-free domain adaptation with CrossEntropyLoss, implementing model consistency regularization to prevent excessive deviation between target and source models. For single-label classification, we select ProDe Tang et al. (2024a) as the baseline. We conducted comprehensive experiments across different tasks, datasets, and architectures, encompassing both source-available and source-free domain adaptation scenarios.

**Semantic Segmentation Methods:** For source-available adaptation, we compared against TransDA-B Chen et al. (2022), DAFormer Hoyer et al. (2022a), HRDA Hoyer et al. (2022b), and IDM Wang et al. (2023). For source-free adaptation, we evaluated against DAFormer Hoyer et al. (2022a), HRDA Hoyer et al. (2022b), IDM Wang et al. (2023), ATP Wang et al. (2024), MISFIT Rizzoli et al. (2024), TENT Wang et al. (2020), CoTTA Wang et al. (2022a), DePT Gan et al. (2023), VDP Gao et al. (2022), SFKT Liu et al. (2021), and SFDA-Seg Kundu et al. (2021).

**Classification Methods:** In single-label image classification, we compared with source-available methods including PDA Bai et al. (2024), DAMP Du et al. (2024), STAR Lu et al. (2020), and RWOT Xu et al. (2020), and source-free methods including SHOT Liang et al. (2020), DIFO Tang et al. (2024b), ProDe Tang et al. (2024a), ATP Wang et al. (2024), PLUE Litrico et al. (2023), and SFDA+ Mitsuzumi et al. (2024).

**Network Architectures:** Our experiments span multiple architectures of varying scales, including Segformer-B0/B1/B2/B3/B4/B5, P2T-Base, ResNet-50/101, and VGG16, demonstrating the generalizability of our approach across different network designs.

**Semantic Segmentation Datasets:** We evaluated on three challenging adaptation scenarios: GTA5→Cityscapes, SYNTHIA→Cityscapes, and Cityscapes→ACDC. The synthetic dataset GTA5 Richter et al. (2016) contains 24,966 annotated images with a resolution of 1914×1052, taken from the famous game Grand Theft Auto with ground truth generated by game rendering. SYNTHIA Ros et al. (2016) is another synthetic dataset containing 9,400 fully annotated images with a resolution of 1280×760. Cityscapes Cordts et al. (2016) consists of 2,975 annotated training images and 500 validation images with a resolution of 2048×1024. The Adverse Conditions Dataset (ACDC) Sakaridis et al. (2021) contains four different adverse visual conditions: Fog, Night, Rain, and Snow, sharing the same semantic classes with Cityscapes. These datasets enable evaluation of source-free adaptation from synthetic to real domains and from normal to adverse conditions.

**Classification Datasets:** For image classification, we selected four datasets of varying scales: Office-31 Saenko et al. (2010) is a small-scale dataset including three domains (Amazon, Webcam, and Dslr) with 4,652 images of 31 categories taken in office environments. Office-Home Venkateswara et al. (2017) is a medium-scale dataset containing 15k images belonging to 65 categories from four domains: Artistic images, Clip Art, Product images, and Real-world images. VisDA-C Peng et al. (2017) is a large-scale dataset with synthetic to real transfer tasks, containing 152k synthetic source images and 55k real target images from Microsoft COCO. DomainNet Saito et al. (2019) is a challenging large-scale dataset created by removing noisy labels from the original version, containing 600k images of 345 classes from 6 domains with varying image styles.

### A.4.3  IMAGE SEGMENTATION EFFECT

We visualize segmentation outputs on GTA5 → Cityscapes in Figure I to qualitatively evaluate SHLSA's effectiveness. Compared to the *Source Only* baseline, SHLSA significantly improves semantic structure and boundary accuracy. It recovers fine details in challenging regions such as traffic participants (*person*, *rider*, *car*) and urban infrastructure (*pole*, *traffic sign*), which are often under-segmented or misclassified by source-only models. These improvements are especially clear in cluttered or occluded scenes where traditional methods tend to suffer from **class confusion** and **unstable decision boundaries**.

These gains come from SHLSA's core components: *stepwise alignment* progressively reduces domain gaps via pseudo-source guidance, addressing semantic misalignment; *hierarchical semantic aggregation* captures multi-scale context for more stable boundaries; and *confidence-aware complementary learning* suppresses noisy predictions and refines uncertain areas. Together, they help SHLSA maintain semantic consistency and achieve precise, fine-grained segmentation under challenging domain shifts.

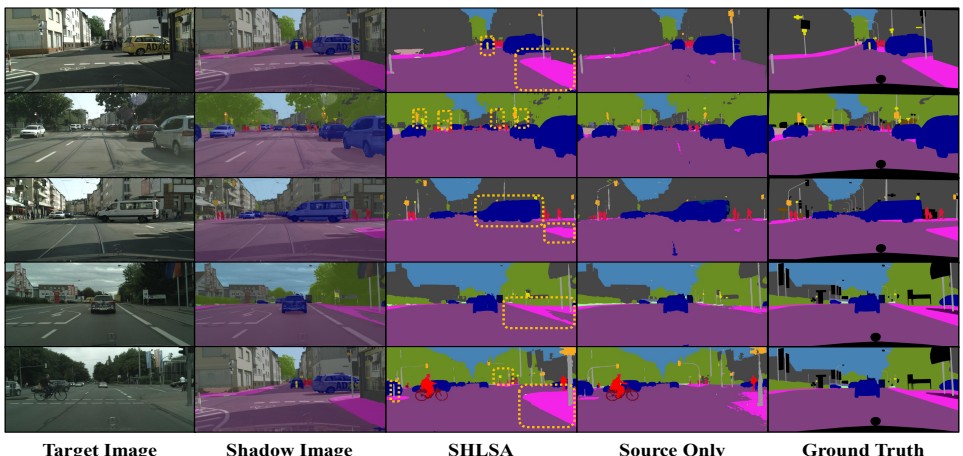

| Target Image | Shadow Image | SHLSA | Source Only | Ground Truth |

Figure I: Visualization for predicted segmentation masks on the GTA5 → Cityscapes task.

### A.4.4  CONFUSION MATRIX FOR IMAGE CLASSIFICATION

To better understand the behavior of SHLSA under different classification scenarios, we conduct a confusion matrix analysis across three datasets in decreasing order of scale and complexity: VisDA-C, Office-Home, and Office-31. This analysis complements the quantitative accuracy metrics by highlighting which classes are most frequently confused and why. It also provides insight into the method's fine-grained discriminative ability and its robustness to inter-class semantic proximity or visual ambiguity under domain shift.

**Confusion Analysis on VisDA-C.** VisDA-C is a challenging synthetic-to-real benchmark comprising over 280K images across 12 object categories. As shown in Figure II, the confusion matrix for the *train→val* task unveils frequent misclassifications among visually similar classes. For instance, *car* is often misclassified as *person* (916 instances), and there is substantial confusion between *bus* and *truck*, as well as between *bicycle* and *motorcycle*. These trends reflect the difficulty of preserving fine-grained semantics under domain shift, especially in real-world scenes characterized by complex textures, occlusions, and noisy backgrounds. While SHLSA successfully mitigates many such confusions through its hierarchical semantic alignment mechanism, its use of fixed-level feature aggregation imposes certain constraints. Specifically, it may struggle in scenarios with high intra-class variability or overlapping inter-class appearances. This suggests that future improvements could benefit from incorporating more flexible, instance-aware representations capable of adapting to subtle semantic distinctions and dynamic visual contexts.

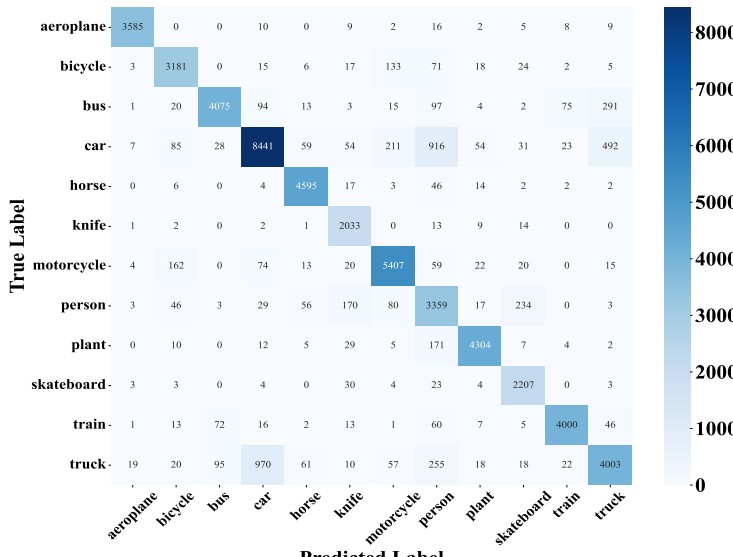

Figure II: Confusion matrices for different domain shifts in the VisDA-C dataset.

**Confusion Analysis on Office-Home.** Office-Home is a medium-scale benchmark spanning 4 domains and 65 categories. As shown in Figure III, confusion in the *Ar→Rw* task is more dispersed due to the dataset's fine-grained label space and higher visual variability. For instance, *monitor* is often misclassified as *computer* (12 times), *bottle* as *soda* (15 times), and *pen* as *marker*, reflecting the challenges in distinguishing semantically related and visually similar objects. While SHLSA mitigates domain shift through hierarchical integration of local and global semantics, its fixed fusion strategy may struggle with class-specific ambiguity, particularly in densely populated and complex label spaces. This highlights the potential benefit of adopting more adaptive aggregation mechanisms that better capture intermediate semantic complexity and enhance discrimination.

**Confusion Analysis on Office-31.** Office-31 is a small-scale dataset with 3 domains and 6 transfer tasks. In the Amazon→DSLR (A→D) setting (Figure IV), the model achieves strong overall performance but still confuses visually or contextually similar classes, such as *monitor* vs. *desk lamp* and *laptop* vs. *ring binder*. These errors likely stem from shared shapes or frequent co-occurrence. While SHLSA handles mild domain shifts well via stepwise alignment, its global fusion may overlook subtle distinctions, suggesting the need for more localized or instance-aware modeling.

**Cross-Dataset Comparison.** The three datasets reflect a spectrum of domain adaptation challenges, from the large-scale synthetic-to-real gap in VisDA-C to the denser label space of Office-Home and the relatively mild shifts in Office-31. SHLSA maintains robust performance across these settings by progressively aligning semantic features, yet residual confusions persist in visually or functionally similar categories. This suggests that while SHLSA effectively addresses broad distribution gaps, its current feature modeling may lack the granularity needed for fine-grained discrimination. Incorporating part-aware cues, adaptive fusion, or relational reasoning could further improve its adaptability in complex or densely labeled domains.

### A.4.5 T-SNE VISUALIZATION FOR IMAGE CLASSIFICATION ON VISDA-C

Figure V illustrates the feature distributions of source-only, SHOT, and SHLSA on the VisDA-C dataset using T-SNE visualization. The **source-only features (a)** show significant overlap and scattered clusters, indicating severe **class confusion** and weak semantic separability under domain shift. This suggests that models trained only on source data struggle to form clear boundaries between classes in the target domain.

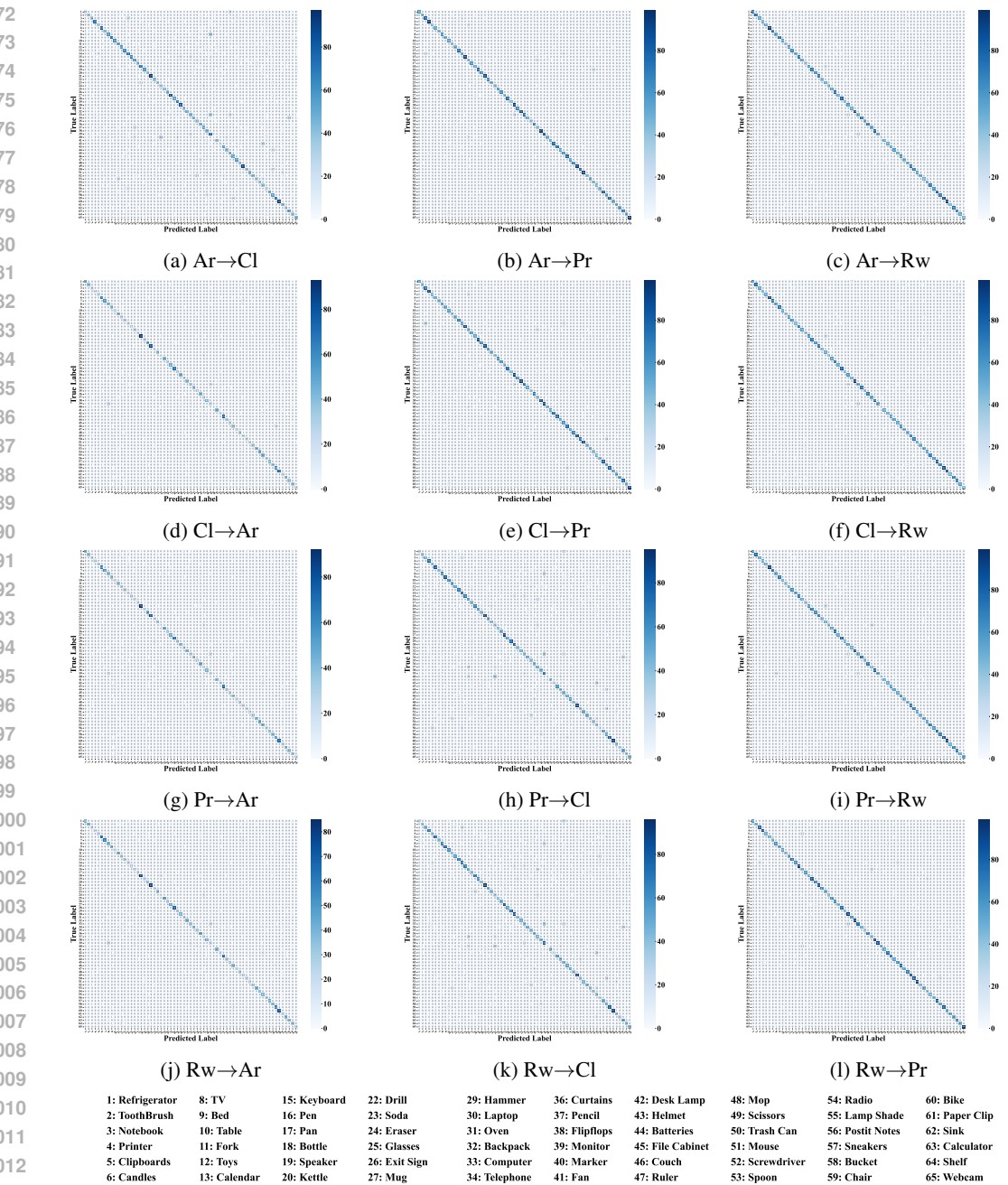

| 1: Refrigerator | 8: TV | 15: Keyboard | 22: Drill | 29: Hammer | 36: Curtains | 42: Desk Lamp | 48: Mop | 54: Radio | 60: Bike |
| 2: ToothBrush | 9: Bed | 16: Pen | 23: Soda | 30: Laptop | 37: Pencil | 43: Helmet | 49: Scissors | 55: Lamp Shade | 61: Paper Clip |
| 3: Notebook | 10: Table | 17: Pan | 24: Eraser | 31: Oven | 38: Flipflops | 44: Batteries | 50: Trash Can | 56: Postit Notes | 62: Sink |
| 4: Printer | 11: Fork | 18: Bottle | 25: Glasses | 32: Backpack | 39: Monitor | 45: File Cabinet | 51: Mouse | 57: Sneakers | 63: Calculator |
| 5: Clipboards | 12: Toys | 19: Speaker | 26: Exit Sign | 33: Computer | 40: Marker | 46: Couch | 52: Screwdriver | 58: Bucket | 64: Shelf |
| 6: Candles | 13: Calendar | 20: Kettle | 27: Mug | 34: Telephone | 41: Fan | 47: Ruler | 53: Spoon | 59: Chair | 65: Webcam |
| 7: Knives | 14: Folder | 21: Flowers | 28: Alarm Clock | 35: Push Pin | | | | | |

Figure III: Confusion matrices for different domain shifts in the Office-Home dataset.

The **SHOT method (b)** partially improves the compactness and separation of clusters. However, some ambiguity and overlap persist among visually or semantically related classes, revealing that SHOT's global alignment alone cannot fully resolve fine-grained category differences.

In contrast, **SHLSA (c)** generates the most distinct and well-separated clusters. Its hierarchical semantic alignment enhances both intra-class cohesion and inter-class separation, effectively reducing class confusion. This results in more stable decision boundaries and improved feature discriminability. The visualization thus provides strong empirical support for SHLSA's ability to handle challenging domain shifts better than conventional low-level alignment methods by capturing richer semantic structure.

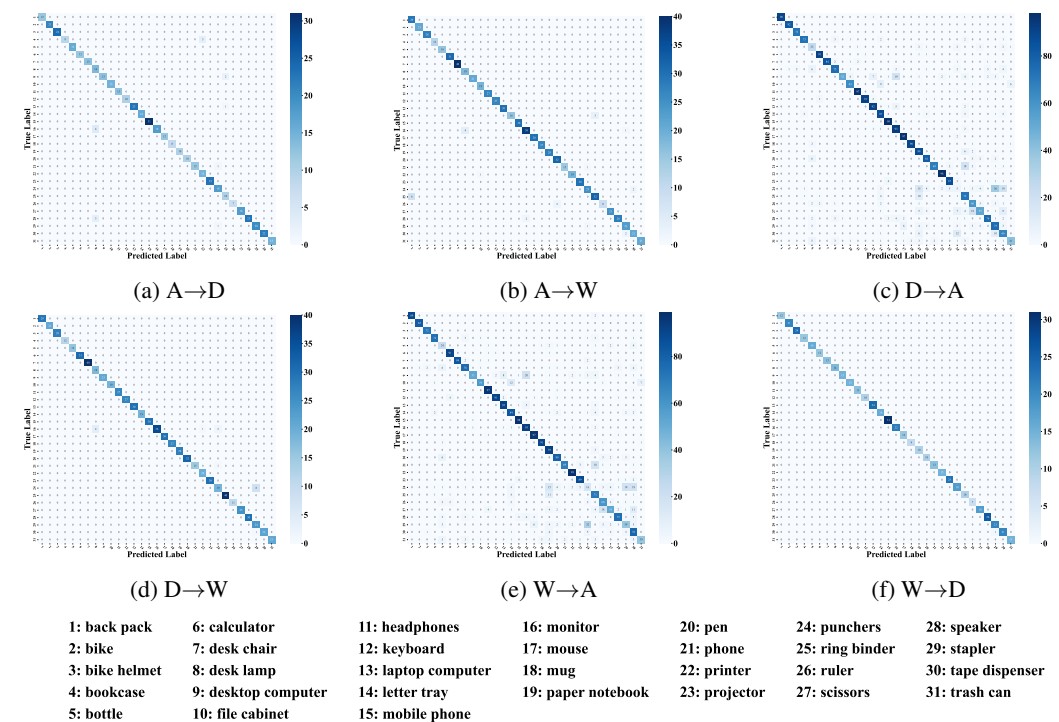

(a) A→D  (b) A→W  (c) D→A

(d) D→W  (e) W→A  (f) W→D

| | | | | | |
|---|---|---|---|---|---|
| 1: back pack | 6: calculator | 11: headphones | 16: monitor | 20: pen | 24: punchers | 28: speaker |
| 2: bike | 7: desk chair | 12: keyboard | 17: mouse | 21: phone | 25: ring binder | 29: stapler |
| 3: bike helmet | 8: desk lamp | 13: laptop computer | 18: mug | 22: printer | 26: ruler | 30: tape dispenser |
| 4: bookcase | 9: desktop computer | 14: letter tray | 19: paper notebook | 23: projector | 27: scissors | 31: trash can |
| 5: bottle | 10: file cabinet | 15: mobile phone | | | | |

Figure IV: Confusion matrices for different domain shifts in the Office-31 dataset.

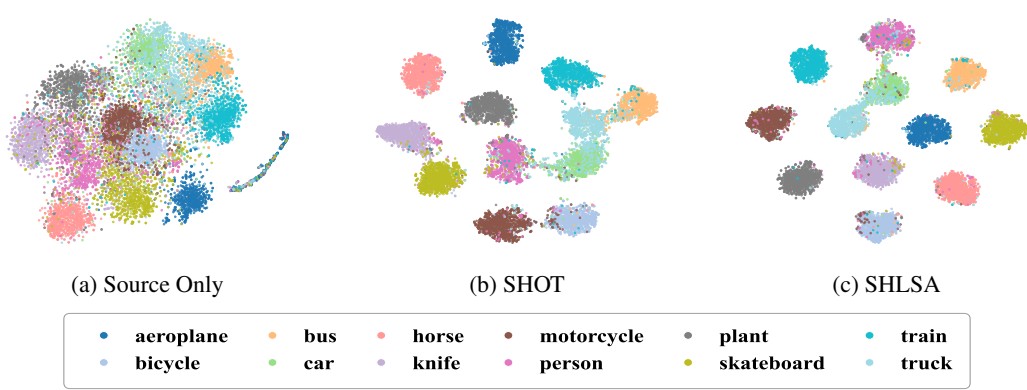

(a) Source Only  (b) SHOT  (c) SHLSA

- aeroplane
- bicycle
- bus
- car
- horse
- knife
- motorcycle
- person
- plant
- skateboard
- train
- truck

Figure V: t-SNE visualization for image classification on VisDA-C dataset.

### A.4.6 CROSS-MODULE DEPENDENCY ANALYSIS

To address concerns regarding insufficient analysis of cross-module dependencies, comprehensive ablation experiments were conducted to analyze the relationships between HFA, CACL, and SDA components. The results are presented in Table I.

The baseline represents source-free domain adaptation using SHOT-based source hypothesis transfer. Configuration (a) modifies the backbone encoder to HFA, (b) adds CACL post-processing to the baseline, (c) performs stepwise alignment on the baseline, (d) removes SDA operations from SHLSA for direct one-step alignment, (e) removes CACL post-processing from SHLSA, (f) replaces the SHLSA encoder with the backbone encoder, and (g) represents the complete SHLSA method.

The experimental results reveal several key insights regarding cross-module dependencies:

Table I: Cross-module Ablation Studies on GTA5→Cityscapes.

| Method | HFA | CACL | SDA | mIoU |
|--------|-----|------|-----|------|
| Baseline | | | | 44.5 |
| (a) | ✓ | | | 57.5 |
| (b) | | ✓ | | 50.2 |
| (c) | | | ✓ | 53.4 |
| (d) | ✓ | ✓ | | 65.6 |
| (e) | ✓ | | ✓ | 66.7 |
| (f) | | ✓ | ✓ | 60.3 |
| (g) | ✓ | ✓ | ✓ | 69.2 |

- **Individual Component Contributions:** From experiments (a), (b), and (c), HFA contributes most significantly to SHLSA performance improvement, followed by SDA, with CACL showing the weakest individual contribution.

- **HFA-CACL Synergy:** Comparing baseline with (b) and (a) with (d), CACL demonstrates higher performance gains when HFA is present (57.5→65.6) compared to without HFA (44.5→50.2). This indicates that HFA enhances semantic feature quality, thereby improving CACL's label quality.

- **HFA-SDA Interaction:** Similarly, comparing baseline with (c) and (a) with (e), SDA shows more pronounced performance gains with HFA (57.5→66.7) than without (44.5→53.4).

- **CACL-SDA Relationship:** Comparing baseline with (b) and (c) with (f), CACL demonstrates more effective performance when SDA is present (44.5→50.2 vs. 53.4→60.3). Conversely, CACL and SDA also promote HFA's performance enhancement (from 44.5→57.5 to 50.2→65.6 and 53.4→66.7). However, CACL's promotion effect on SDA is less pronounced (44.5→53.4 vs. 50.2→60.3).

- **Diminishing Returns:** Under conditions of HFA+CACL, CACL+SDA, or HFA+SDA, the performance gains of the remaining modules are relatively lower compared to implementing only one module or the baseline. This phenomenon may be attributed to diminishing marginal effects as segmentation performance improves.

These findings demonstrate that the three components exhibit strong interdependencies, with HFA serving as the foundation for enhancing the effectiveness of both CACL and SDA, while CACL and SDA provide complementary improvements that collectively contribute to the superior performance of the complete SHLSA framework.

### A.4.7 ONLINE TEST-TIME ADAPTATION EVALUATION

To address concerns regarding limited experimental scope and lack of evaluation on challenging temporal scenarios, additional experiments were conducted to evaluate SHLSA's performance in online test-time adaptation settings. This evaluation extends beyond the large-scale single-image classification experiments on DomainNet to include temporal adaptation scenarios.

**Experimental Setup**  The online test-time adaptation setting differs significantly from traditional offline scenarios. Specifically, the batch size was adjusted to appropriate mini-batches (e.g., from 64 to 128), and the testing process was modified to ensure streaming data processing. In a single epoch, the model outputs results immediately after training on each mini-batch, ensuring that data flows in a streaming manner where each sample participates in gradient updates only once. This contrasts with offline scenarios where each sample participates in multiple gradient updates and testing occurs only after complete model training.

**Results and Analysis**  Table II presents the performance of SHLSA in online scenarios on Office-Home and VisDA-C datasets. The results demonstrate SHLSA's capability to maintain competitive

performance even under the constraints of online adaptation, where the model must adapt continuously to new data without the benefit of multiple training epochs.

Table II: Performance of SHLSA in Online Scenarios on Image Classification.

| Method | Office-Home | VisDA-C | Avg. |
|---|---|---|---|
| TENT | 67.27 | 75.43 | 71.35 |
| SHOT | 67.89 | 76.69 | 72.29 |
| TENT | 76.62 | 82.58 | 79.60 |

The experimental results indicate that SHLSA maintains robust performance in online adaptation scenarios. For Office-Home, the method achieves an average accuracy of 76.62% across all domain transfer tasks, while for VisDA-C, it achieves 82.58% accuracy. These results are particularly noteworthy given the challenging nature of online adaptation, where the model must adapt incrementally without access to the complete target dataset.

**Temporal Adaptation Characteristics**  The online test-time adaptation results demonstrate several key characteristics of SHLSA's temporal adaptation capabilities:

- **Incremental Learning:** SHLSA effectively adapts to new target domain samples in a streaming fashion, maintaining performance stability without catastrophic forgetting.
- **Efficiency:** The method's ability to perform well with single-pass gradient updates indicates computational efficiency suitable for real-time applications.
- **Robustness:** The consistent performance across different domain transfer scenarios suggests that SHLSA's hierarchical feature alignment and stepwise domain alignment mechanisms are robust to temporal variations in data distribution.

These findings extend SHLSA's applicability to practical scenarios where continuous adaptation to evolving data distributions is required, such as autonomous driving systems or real-time video analysis applications.

## A.5  Parameter Sensitivity Analysis and Mitigation

### A.5.1  Sensitivity of key parameters.

We conduct comprehensive sensitivity analysis on several key parameters in SHLSA, including the positive threshold $\tau_{pos}$, the negative threshold $\tau_{neg}$, the split ratio $\tau_{par}$, and the local-global hyperparameters $\tau_{lg}$ (ratio) and $\tau_{lw}$ (local window size).

**Positive threshold $\tau_{pos}$** serves as a filter for probability distributions, treating labels with probabilities above $\tau_{pos}$ as positive while ignoring others. As demonstrated in Fig. 2(b), $\tau_{pos}$ exhibits relatively low sensitivity, primarily because it only influences the inclusion or exclusion of a limited number of positive labels without substantially affecting the overall learning dynamics.

**Negative threshold $\tau_{neg}$** emerges as the most sensitive parameter, critically controlling the distinction between *likely correct* and *unlikely correct* predictions. Within the CACL framework, predictions undergo descending probability sorting, with entries exceeding the gradient threshold $\tau_{neg}$ being designated as negative pseudo-labels. This heightened sensitivity stems from several interconnected factors: (1) $\tau_{neg}$ directly governs negative pseudo-label quality within CACL; (2) During early training phases characterized by high model uncertainty and relatively flat probability distributions, minor parameter adjustments can dramatically alter negative label quantities; (3) Excessively low $\tau_{neg}$ values introduce erroneous negative labels, causing noise accumulation and performance degradation, which is the primary source of error propagation compared to $\tau_{pos}$.

**Split ratio $\tau_{par}$** regulates the proportion of target domain data allocated to the pseudo-source domain within SDA. For tasks exhibiting substantial domain shifts (e.g., VisDA-C), employing fewer low-entropy samples as pseudo-source domain through reduced $\tau_{par}$ values enhances sample quality. Conversely, for tasks with moderate shifts (e.g., Office-31), elevated $\tau_{par}$ values enable better

exploitation of source-like samples within the target domain. Table III validates that confidence-aware dynamic $\tau_{par}$ configuration consistently outperforms fixed threshold approaches across diverse tasks. Specifically, we followed the idea of CACL and selected the points with large variations in self-entropy of the samples in the target domain as the partition points.

Table III: Impact of Different $\tau_{par}$ Parameters on Single-label Classification

| Method | $\tau_{par}^{-}$ | Office-31 (0.1335) | Office-Home (0.2491) | VisDA-C (0.1677) | DomainNet (0.3478) |
|---|---|---|---|---|---|
| Conf-Aware | 0.92 | 92.8 | - | - | - |
| | 0.87 | - | 85.0 | - | - |
| | 0.79 | - | - | 91.3 | - |
| | 0.44 | - | - | - | 83.7 |
| Fixed | 0.10 | 92.5 | 84.7 | 90.9 | 83.4 |
| | 0.20 | 92.5 | 84.6 | 91.2 | 83.5 |
| | 0.30 | 92.6 | 84.7 | 91.2 | 83.5 |
| | 0.40 | 92.5 | 84.7 | 91.1 | 83.7 |
| | 0.50 | 92.7 | 84.8 | 91.3 | 83.7 |
| | 0.60 | 92.6 | 84.9 | 91.1 | 83.5 |
| | 0.70 | 92.8 | 85.0 | 91.3 | 83.3 |
| | 0.80 | 92.8 | 85.0 | 91.3 | 83.1 |
| | 0.90 | 92.8 | 84.9 | 91.2 | 82.9 |

**Local–global ratio $\tau_{lg}$ and local window size $\tau_{lw}$** demonstrate distinct sensitivity patterns across different tasks and window configurations. As shown in Table IV, segmentation tasks (GTA5→Cityscapes) exhibit significantly higher parameter sensitivity compared to classification tasks (VisDA-C). For segmentation tasks, sensitivity varies considerably with window size: large windows (1024 × 1024) show minimal sensitivity to $\tau_{lg}$ with performance variations of only 0.4% (68.8-69.2 mIoU), while smaller windows demonstrate progressively increased sensitivity, where 512×512 windows show 0.7% variation (68.4-69.1 mIoU) and 256×256 windows exhibit the highest sensitivity with 1.3% variation (67.9-68.8 mIoU). This pattern indicates that smaller windows lack sufficient semantic context and become more dependent on the precise balance between local and global information. In contrast, classification tasks (VisDA-C) demonstrate remarkably stable performance across all parameter combinations, with minimal variations of 0.2% for 256×256 windows (91.1-91.3 mAcc) and 0.8% for 128×128 windows (90.6-91.4 mAcc). This low sensitivity reflects that classification tasks with sparse semantic structures are less dependent on fine-grained local-global feature alignment and primarily rely on global semantic representations.

Table IV: Sensitivity analysis of $\tau_{lg}$ with varying $\tau_{lw}$ on different datasets.

| $\tau_{lg}/ \tau_{lw}$ | GTA5 → Cityscapes | | | VisDA-C | |
|---|---|---|---|---|---|
| | 1024×1024 | 512×512 | 256×256 | 256×256 | 128×128 |
| 0.1 | 68.8 | 68.9 | 68.1 | 91.1 | 91.3 |
| 0.2 | 68.9 | 68.8 | 68.1 | 91.1 | 91.4 |
| 0.3 | 69.1 | 68.9 | 68.2 | 91.2 | 90.9 |
| 0.4 | 69.0 | 68.7 | 68.5 | 91.1 | 91.0 |
| 0.5 | 69.2 | 69.1 | 68.7 | 91.3 | 91.1 |
| 0.6 | 68.8 | 68.9 | 68.8 | 91.2 | 90.9 |
| 0.7 | 68.8 | 68.6 | 68.4 | 91.3 | 90.7 |
| 0.8 | 68.8 | 68.5 | 68.3 | 91.2 | 90.8 |
| 0.9 | 68.8 | 68.4 | 67.9 | 91.2 | 90.6 |

### A.5.2 MITIGATION FOR HIGHLY SENSITIVE PARAMETERS.

For parameters with low sensitivity ($\tau_{pos}$, $\tau_{par}$, $\tau_{lg}$, $\tau_{lw}$), fixed values can be set based on empirical results. For the highly sensitive parameter $\tau_{neg}$, we propose adaptive strategies to mitigate error accumulation: **Conservative initialization** sets $\tau_{neg}$ to a high value initially to prevent large fluctuations; **Early-stage restriction** constrains $\tau_{neg}$ to a narrow range in early epochs; **Dynamic adjustment** adapts $\tau_{neg}$ based on gradient stability; **Uncertainty-aware tuning** uses entropy metrics to balance precision and recall.

We evaluate three implementations: **Strategy 1** fixes $\tau_{\text{neg}} = 0.9$ for the first 5 epochs; **Strategy 2** keeps $\tau_{\text{neg}} \in [0.6, 0.9]$ initially then expands to $[0.4, 0.9]$; **Strategy 3** dynamically adjusts $\tau_{\text{neg}}$ every two epochs based on gradient variance. Table V shows that these dynamic strategies effectively mitigate sensitivity and improve SHLSA's stability and performance.

Table V: Performance comparison of mitigation strategies for $\tau_{\text{neg}}$ across epochs.

| Epoch | Baseline | Strategy 1 | Strategy 2 | Strategy 3 |
|-------|----------|------------|------------|------------|
| 1 | 88.74 | 88.90 | 89.00 | 89.20 |
| 2 | 89.73 | 89.91 | 90.00 | 90.10 |
| 3 | 90.06 | 90.15 | 90.20 | 90.30 |
| 4 | 90.20 | 90.31 | 90.35 | 90.50 |
| 5 | 90.34 | 90.47 | 90.50 | 90.70 |
| 6 | 90.51 | 90.63 | 90.65 | 90.90 |
| 7 | 90.75 | 90.81 | 90.83 | 91.05 |
| 8 | 90.54 | 90.70 | 90.95 | 91.20 |
| 9 | 90.63 | 90.85 | 91.00 | 91.25 |
| 10 | 90.76 | 90.80 | 91.08 | 91.21 |

### A.5.3 THEORETICAL GUIDANCE FOR PARAMETER SELECTION

**Bridging Theory and Practice**  Theorem 1 establishes the theoretical foundation for threshold selection by proving that when the entropy of output probability distributions is sufficiently low ($\mathcal{H}(p) \leq H_0$), there exist two thresholds $\tau_\alpha > \tau_\beta$ such that the high-confidence positive set $\mathcal{Y}_+$ and low-confidence negative set $\mathcal{Y}_-$ can be well-separated in expectation, with the probability mass of $\mathcal{Y}_-$ being strictly controlled to prevent excessive noise from negative pseudo-labels.

While Theorem 1 does not directly specify the experimental parameters $\tau_{\text{pos}}$ and $\tau_{\text{neg}}$, it provides theoretical guidance for their selection. The correspondence between theoretical and practical parameters is established as follows: $\tau_\alpha = \tau_{\text{pos}}$ represents the absolute probability threshold for positive pseudo-labels, while $\tau_\beta$ corresponds to the probability of the first category satisfying the gradient condition, defined as:

$$\tau_\beta = p_{\arg\max_i\{|p_i^* - p_{i-1}^*|\} \geq \tau_{\text{neg}}}, \tag{27}$$

where $p_i^*$ denotes the probability of the $i$-th category after descending sorting.

**Derivation of Threshold Bounds**  Given a target distribution $\mathbf{p} = (p_1, \ldots, p_C)$ with entropy $\mathcal{H}(\mathbf{p}) \leq H_0$, we define $G(t) = \sum_{c:p_c \leq t} p_c$ as the cumulative probability mass below threshold $t$. To ensure the total probability mass of the negative set does not exceed $\epsilon > 0$, we require $G(\tau_\beta) \leq \epsilon$.

From Theorem 1's inequality, we derive the upper bound:

$$\tau_\beta \leq \exp\left(-\frac{H_0}{\epsilon}\right). \tag{28}$$

This constrains the feasible range of $\tau_\beta$ to $(0, \exp(-H_0/\epsilon))$. Smaller tolerance $\epsilon$ results in tighter upper bounds for $\tau_\beta$.

For practical implementation with $H_0 = 0.5$ and tolerance $\epsilon = 0.05$ (5%), we obtain:

$$\tau_\beta \leq \exp\left(-\frac{0.5}{0.05}\right) = \exp(-10) \approx 4.5 \times 10^{-5}. \tag{29}$$

**Practical Parameter Derivation**  To derive $\tau_{\text{neg}}$ from the theoretical bounds, we control the quality of tail negative pseudo-labels using $\sum_{i>k} p_i \leq \epsilon$. Given the gradient definition $p_{k+1} \leq p_k - \tau_{\text{neg}}$ and assuming the worst-case scenario where $p_{k+1}$ is the maximum in the tail:

$$\sum_{i>k} p_i \leq (C-k)p_{k+1} \leq (C-k)(p_k - \tau_{\text{neg}}). \tag{30}$$

With $\tau_\alpha = \tau_{\text{pos}} = 0.9$, we have:

$$\sum_{i>k} p_i \leq (C - k)(0.9 - \tau_{\text{neg}}). \tag{31}$$

To ensure the right side satisfies $\leq \epsilon$:

$$0.9 - \tau_{\text{neg}} \leq \frac{\epsilon}{C - k} \Rightarrow \tau_{\text{neg}} \geq 0.9 - \frac{\epsilon}{C - k}. \tag{32}$$

For typical SHLSA scenarios with $C \approx 10$, $k > 1$, and $\epsilon = 0.05$:

$$\tau_{\text{neg}} \geq 0.9 - \frac{0.05}{10 - 1} \approx 0.9 - 0.0056 \approx 0.8944. \tag{33}$$

Therefore, setting $\tau_{\text{pos}} = 0.9$ and controlling negative pseudo-label quality error to $\epsilon \leq 0.05$ yields $\tau_{\text{neg}} \approx 0.9$, which aligns with our experimental findings in the parameter sensitivity analysis.

**Handling High-Entropy Samples** Theorem 1 assumes $\mathcal{H}(p) \leq H_0$. When significant distribution shifts occur in the target domain, high-entropy samples become prevalent, making $\mathcal{H}(p) \approx H_0$ and potentially violating the theoretical assumptions. However, CACL maintains pseudo-label reliability through several mechanisms:

- **Positive Pseudo-label Filtering:** The absolute probability threshold $\tau_{\text{pos}}$ automatically filters high-entropy samples. Positive pseudo-labels are accepted only when $p_i \geq \tau_{\text{pos}}$ (e.g., 0.9), while high-entropy samples typically have maximum probabilities well below 0.9 and are thus rejected. This ensures positive pseudo-labels are extracted only from low-entropy, confident samples.

- **Stable Negative Pseudo-label Selection:** Negative pseudo-labels rely on probability gradient ratios $\tau_{\text{neg}}$, which are more stable than absolute probabilities. For high-entropy samples with moderate maximum probabilities, tail categories still exhibit clear decreases. High entropy typically results from multiple possible categories with several high $p_i$ values, but tail probability differences remain more stable than global entropy, enabling reliable negative pseudo-label generation.

- **Automatic Weight Reduction:** CACL automatically reduces the influence of high-entropy samples during training. Both contrastive loss and pseudo-label loss depend on prediction probabilities, meaning high-entropy samples with flatter probability distributions contribute smaller gradients (based on CrossEntropyLoss). Low-entropy confident samples provide stronger supervision through pseudo-labels. This mechanism ensures that erroneous high-entropy pseudo-labels do not destabilize training, as reliable low-entropy pseudo-labels dominate gradient updates, guaranteeing training stability.

## A.6 COMPUTATIONAL COST ANALYSIS

To comprehensively evaluate the computational efficiency of SHLSA, we conduct both qualitative and quantitative analyses of the resource requirements. We first analyze the theoretical complexity of each component, then perform empirical evaluations across different architectures, and finally explore optimization strategies.

### A.6.1 QUALITATIVE ANALYSIS OF COMPUTATIONAL COMPLEXITY

To identify the factors contributing to the increased computational cost of SHLSA, we analyze its framework, which consists of *Stepwise Domain Alignment* (SDA), *Hierarchical Feature Alignment* (HFA), and *Contrastive Adaptive Confidence Learning* (CACL). The CACL component, being a post-processing method, requires only $O(n \log n)$ sorting and $O(n)$ first-order gradient computation, thus introducing minimal computational overhead.

The computational overhead primarily arises from the SDA and HFA components. In the SDA stage, *pseudo-source domain* samples undergo two passes through the HFA module: the first pass aligns with common semantics before mixing, and the second pass guides the *remaining target domain*

post-mixing. Consequently, the main computational cost increase is attributed to the MixMatch operations in SDA and the design of the HFA module.

For the MixMatch operations, the time complexity is determined by the number of data augmentations $K$, the batch size of labeled data (*pseudo-source domain*) $L$, and the batch size of unlabeled data (*remaining target domain*) $U$. Thus, the overall time complexity is $O(K \cdot L \cdot U)$.

For the HFA module, the complexity arises from two parts: the feature encoder and the attention module.

**Feature Encoder.** The feature encoder processes local and global information across different abstraction levels to capture semantics at different granularities. The time complexity impact is similar to MobileViT Mehta & Rastegari (2021)'s local attention, reducing the original $O(n^2)$ operations to $O\left(k(\frac{n}{k})^2\right) = O\left(\frac{n^2}{k}\right)$. However, HFA additionally computes global attention to obtain coarse-grained semantic information, resulting in a complexity of:

$$O\left((k+1)(\frac{n}{k})^2\right) = O\left(\frac{(k+1) \cdot n^2}{k^2}\right). \tag{34}$$

Here, $k$ denotes the number of local patches, and global semantic information is obtained by resizing the original image to local size, thus having a complexity of $O\left(\left(\frac{n}{k}\right)^2\right)$. In practice, $k \in [4, 9]$, ensuring $O\left(\frac{k+1}{k^2} \cdot n^2\right) < O(n^2)$. The main computational cost increase occurs in the subsequent attention fusion operations.

**Attention Module.** This module performs attention fusion on $(\frac{n}{k})^2$ pixels across $k+1$ patches, resulting in a complexity of $O\left(\frac{(k+1)^2}{k^2} \cdot n^2\right)$. Notably, this computational overhead is significant in semantic segmentation tasks. For single-image classification tasks, pixel-level attention scores for $k+1$ patches are not required, reducing the complexity to $O((k+1)^2)$, which is manageable under the condition $k \in [4, 9]$. Combining the complexities of the feature encoder and attention module, the overall time complexity of HFA is:

$$O\left(\frac{k+1}{k^2} \cdot n^2 + \frac{(k+1)^2}{k^2} \cdot n^2\right) = O\left(\frac{(k+1)(k+2)}{k^2} \cdot n^2\right) > O(n^2). \tag{35}$$

The computational cost increase is primarily due to the attention fusion process across different semantic levels in semantic segmentation tasks.

### A.6.2 QUANTITATIVE EXPERIMENTS ON COMPUTATIONAL COST

We conduct comprehensive experiments to quantify the computational impact of different components. All experiments are performed on a single NVIDIA GeForce RTX 3090 (24GB) GPU. For semantic segmentation, we evaluate on SYNTHIA→Cityscapes with batch size 2, local window size $\tau_{lw} = 1024 \times 1024$, and 40,000 iterations. For classification, we use Office-Home with batch size 64, $\tau_{lw} = 256 \times 256$, and 10 epochs.

Table VI: Performance and Computational Cost Analysis on Semantic Segmentation.

| Method | GPU Memory (GB) | Time-Item (s/item) | Time-Train (GPU·h) | mIoU (%) | Backbone | Model Size (MB) |
|---|---|---|---|---|---|---|
| SHLSA | 22.4 | 2.40 | 26.7 | 64.1 | Segformer-B5 | 313.1 |
| SHLSA w/o CACL | 22.4 | 2.40 | 26.4 | 63.4 | Segformer-B5 | 313.1 |
| SHLSA w/o HFA | 20.2 | 1.52 | 18.8 | 61.3 | Segformer-B5 | 313.1 |
| SHLSA w/o SDA | 19.9 | 2.40 | 14.3 | 62.7 | Segformer-B5 | 313.1 |
| SHLSA-B4 | 20.4 | 2.12 | 22.1 | 63.5 | Segformer-B4 | 234.4 |
| SHLSA-B3 | 17.0 | 1.84 | 17.4 | 62.3 | Segformer-B3 | 170.3 |
| SHLSA-B2 | 13.1 | 1.66 | 12.4 | 61.2 | Segformer-B2 | 94.4 |
| SHLSA-B1 | 12.6 | 1.50 | 11.7 | 54.3 | Segformer-B1 | 52.2 |
| SHLSA-B0 | 11.7 | 1.47 | 11.2 | 48.9 | Segformer-B0 | 13.7 |
| SHLSA-P2T | 12.3 | 1.49 | 11.5 | 60.2 | P2T-Base | 144.5 |
| SHLSA-ResNet | 13.2 | 1.55 | 11.3 | 58.2 | ResNet-101 | 178.9 |
| ATP | 20.1 | 1.56 | 19.6 | 63.7 | Segformer-B5 | 313.1 |
| ATP-ResNet | 12.5 | 1.51 | 11.0 | 57.6 | ResNet-101 | 178.9 |
| ATP-P2T | 11.9 | 1.32 | 9.7 | 59.6 | P2T-Base | 144.5 |
| SFKT | 10.8 | 1.25 | 8.3 | 45.9 | ResNet-101 | 178.9 |
| SFDA-Seg | 11.3 | 1.27 | 8.5 | 48.9 | ResNet-101 | 178.9 |

Table VII: Performance and Computational Cost Analysis on Classification.

| Method | GPU Memory (GB) | Time-Item (s/item) | Time-Train (GPU·h) | mAcc (%) | Backbone | Model Size (MB) |
|---|---|---|---|---|---|---|
| SHLSA | 10.8 | 0.65 | 0.17 | 85.0 | ResNet-50 | 102.6 |
| SHLSA w/o CACL | 10.8 | 0.65 | 0.16 | 84.9 | ResNet-50 | 102.6 |
| SHLSA w/o HFA | 9.7 | 0.57 | 0.15 | 84.6 | ResNet-50 | 102.6 |
| SHLSA w/o SDA | 8.2 | 0.65 | 0.13 | 84.7 | ResNet-50 | 102.6 |
| SHLSA-RN101 | 13.6 | 0.67 | 0.18 | 89.3 | ResNet-101 | 178.9 |
| SHLSA-VGG16 | 13.9 | 0.67 | 0.18 | 86.7 | VGG16 | 533.4 |
| SHOT | 7.6 | 0.52 | 0.12 | 71.8 | ResNet-50 | 102.6 |
| SHOT-RN101 | 10.4 | 0.55 | 0.14 | 78.9 | ResNet-101 | 178.9 |
| SHOT-VGG16 | 10.6 | 0.56 | 0.15 | 76.3 | VGG16 | 533.4 |
| TENT | 6.9 | 0.49 | 0.11 | 61.7 | ResNet-50 | 102.6 |
| TENT-RN101 | 9.7 | 0.52 | 0.13 | 70.2 | ResNet-101 | 178.9 |
| TENT-VGG16 | 10.2 | 0.54 | 0.14 | 68.3 | VGG16 | 533.4 |
| SFDA+ | 7.9 | 0.55 | 0.14 | 73.4 | ResNet-50 | 102.6 |

The computational cost analysis provides insights into SHLSA's efficiency characteristics across different components and architectures. Here, SHLSA represents our proposed method, while SHLSA w/o CACL/HFA/SDA denote variants excluding respective components. CACL operates as post-processing during training without affecting inference, HFA involves architectural modifications impacting both training and inference, and SDA is a training methodology affecting only the training process.

**Component-wise Impact.** From Table VI, HFA contributes most significantly to computational overhead. Comparing SHLSA (22.4 GB, 26.7 GPU hours) with SHLSA w/o HFA (20.2 GB, 18.8 GPU hours), HFA increases memory usage by 10.9% and training time by 42.0% due to pixel-level attention fusion requirements. SDA also substantially impacts training efficiency, with SHLSA w/o SDA requiring only 14.3 GPU hours compared to 26.7 hours for the full method (46.4% reduction). CACL shows minimal computational impact, with negligible differences in memory (22.4 GB) and training time (26.4 vs. 26.7 GPU hours).

**Architecture Scalability.** Tables VI and VII demonstrate clear scalability trends. In segmentation, Segformer variants show progressive resource requirements: B0 (11.7 GB, 11.2 GPU hours, 48.9% mIoU) to B5 (22.4 GB, 26.7 GPU hours, 64.1% mIoU). Alternative architectures like P2T-Base (12.3 GB, 11.5 GPU hours, 60.2% mIoU) and ResNet-101 (13.2 GB, 11.3 GPU hours, 58.2% mIoU) offer different performance-efficiency trade-offs. For classification, SHLSA with ResNet-101 achieves higher accuracy (89.3% vs. 85.0%) at increased cost (13.6 GB vs. 10.8 GB).

### A.6.3   OPTIMIZATION STRATEGIES

Based on the computational analysis, three optimization strategies are proposed to improve the efficiency of SHLSA while maintaining competitive performance:

**Lightweight Attention Mechanisms.** Analysis reveals that the attention fusion operations in HFA are the primary cause of significant computational overhead increases. Therefore, employing more lightweight attention mechanisms can effectively reduce SHLSA's computational cost. As demonstrated in Table VIII, replacing the attention mechanism in HFA with more efficient MobileViT operations reduces GPU memory from 22.4GB to 21.2GB while maintaining comparable performance (64.1% vs 63.9% mIoU for semantic segmentation tasks).

**Simplifying SDA's Two-Step Alignment.** When domain shift is relatively small, direct alignment between the pseudo-source domain and remaining target domain can reduce HFA usage by one step while avoiding PRE-NET storage and computation, theoretically improving training efficiency. We propose using self-entropy as an indicator for domain shift assessment.

As shown in Table VIII, when the average entropy is below 0.5 during source model testing on the target domain, simplifying SDA's two-step alignment can improve training efficiency without significantly affecting model performance. For semantic segmentation tasks with dense semantics (SYNTHIA→Cityscapes: 0.5396, GTA5→Cityscapes: 0.7239), removing SDA results in substantial performance degradation (1.4% and 3.6% mIoU drop respectively). In contrast, for image classification tasks with sparse semantics (entropy ¡ 0.35), removing SDA provides significant computational savings (reducing GPU memory from 10.8GB to 8.2GB) with minimal performance impact (¡

1.2% accuracy drop). This suggests that the entropy threshold of 0.5 serves as an effective criterion for determining when to simplify SDA operations.

**Simplifying HFA for Classification Tasks.** For single-label classification tasks with sparse semantics, HFA's hierarchical feature alignment provides limited benefits as these tasks primarily rely on global semantic features rather than fine-grained local patterns. As demonstrated in Table VIII, removing HFA from classification tasks reduces GPU memory usage by 24% (from 10.8GB to 8.2GB) and training time by 20-25% while maintaining comparable performance. The performance drops are minimal: 0.2% for Office-31, 0.3% for Office-Home, 1.0% for VisDA-C, and 1.7% for DomainNet. This suggests that for classification tasks, especially those with low semantic complexity, HFA can be safely omitted to achieve better computational efficiency. This suggests that the entropy threshold of 0.5 serves as an effective criterion for determining when to simplify SDA operations, while HFA simplification is recommended for all single-label classification tasks to balance performance and efficiency.

Table VIII: Performance Optimization Results Across Different Tasks.

| Method | GPU Memory (GB) | Time-Train (GPU·h) | mIoU/mAcc (%) | Task | Dataset | Backbone |
|---|---|---|---|---|---|---|
| SHLSA | 22.4 | 26.5 | 64.1 | | SYNTHIA→Cityscapes (0.5396) | |
| SHLSA+MobileViT | 21.2 | 24.2 | 63.9 | | SYNTHIA→Cityscapes (0.5396) | |
| SHLSA w/o SDA | 19.9 | 14.5 | 62.7 | Segmentation | SYNTHIA→Cityscapes (0.5396) | Segformer-B5 |
| SHLSA | 22.4 | 26.7 | 69.2 | | GTA5→Cityscapes (0.7239) | |
| SHLSA+MobileViT | 21.2 | 24.3 | 68.9 | | GTA5→Cityscapes (0.7239) | |
| SHLSA w/o SDA | 19.9 | 14.3 | 65.6 | | GTA5→Cityscapes (0.7239) | |
| SHLSA | 10.8 | 0.06 | 92.8 | | Office-31 (0.1335) | |
| SHLSA w/o HFA | 8.2 | 0.04 | 92.6 | | Office-31 (0.1335) | |
| SHLSA w/o SDA | 8.2 | 0.04 | 92.7 | | Office-31 (0.1335) | |
| SHLSA | 10.8 | 0.17 | 85.0 | Classification | Office-Home (0.2491) | ResNet-50 |
| SHLSA w/o HFA | 8.2 | 0.13 | 84.7 | | Office-Home (0.2491) | |
| SHLSA w/o SDA | 8.2 | 0.13 | 84.7 | | Office-Home (0.2491) | |
| SHLSA | 10.8 | 1.72 | 91.3 | | VisDA-C (0.1677) | |
| SHLSA w/o HFA | 8.2 | 1.36 | 90.3 | | VisDA-C (0.1677) | |
| SHLSA w/o SDA | 8.2 | 1.36 | 90.6 | | VisDA-C (0.1677) | |
| SHLSA | 10.8 | 1.98 | 83.1 | | DomainNet (0.3478) | |
| SHLSA w/o HFA | 8.2 | 1.53 | 81.4 | | DomainNet (0.3478) | |
| SHLSA w/o SDA | 8.2 | 1.53 | 81.9 | | DomainNet (0.3478) | |

### A.6.4 COMPUTATIONAL COST VS PERFORMANCE TRADE-OFF.

The balance between computational cost and performance is critical in TTA tasks. Table IX presents a comprehensive trade-off analysis using various Segformer architectures (B0-B5), demonstrating an inverse relationship between computational cost and segmentation performance. Considering diminishing performance gains as computational cost increases, Segformer-B2 represents a balanced choice.

SHLSA achieves a favorable trade-off between efficiency and performance. For classification tasks with sparse semantics, removing SDA provides significant computational savings (from 10.8GB to 8.2GB GPU memory) with minimal performance degradation. For semantic segmentation tasks with dense semantics, lightweight attention mechanisms offer a balanced solution. While SHLSA requires more resources than lightweight methods like TENT, the significant performance improvements justify the additional computational cost for accuracy-prioritized applications.

Table IX: Trade-off Analysis Between Computational Cost and Segmentation Performance

| Method | GPU Memory (GB) | Time-Item (s/item) | Time-Train (GPU·h) | mIoU (%) | Backbone |
|---|---|---|---|---|---|
| SHLSA-B0 | 11.7 (+0.0) | 1.47 (+0.00) | 11.2 (+0.0) | 48.9 (+0.0) | Segformer-B0 |
| SHLSA-B1 | 12.6 (+0.9) | 1.50 (+0.03) | 11.7 (+0.5) | 54.3 (+5.4) | Segformer-B1 |
| SHLSA-RN101 | 13.2 (+1.5) | 1.55 (+0.08) | 11.3 (+0.1) | 58.2 (+9.3) | ResNet-101 |
| SHLSA-P2T | 12.3 (+0.6) | 1.49 (+0.02) | 11.5 (+0.3) | 60.2 (+11.3) | P2T-Base |
| SHLSA-B2 | 13.1 (+1.4) | 1.66 (+0.09) | 12.4 (+1.2) | 61.2 (+12.3) | Segformer-B2 |
| SHLSA-B3 | 17.0 (+5.3) | 1.84 (+0.37) | 17.4 (+6.2) | 62.3 (+13.4) | Segformer-B3 |
| SHLSA-B4 | 20.4 (+8.7) | 2.12 (+0.65) | 22.1 (+10.9) | 63.5 (+14.6) | Segformer-B4 |
| SHLSA-B5 | 22.4 (+10.7) | 2.40 (+0.93) | 26.7 (+15.5) | 64.1 (+15.2) | Segformer-B5 |

## A.7 Scaling Effects of SHLSA on Complex Semantic Datasets

This section presents an evaluation of SHLSA's performance on larger and more challenging datasets, focusing on its scaling effects in complex semantic environments.

### A.7.1 Single-Image Classification Performance

The evaluation includes the DomainNet dataset, which is significantly larger and more diverse than previously tested datasets. DomainNet comprises 600k images across 345 categories, providing a robust testbed for assessing SHLSA's adaptability to complex semantic tasks. As shown in Table X, SHLSA demonstrates notable performance improvements over other methods, highlighting its effectiveness in handling extensive semantic categories.

To further investigate the scaling effects of SHLSA, its performance was evaluated on datasets with varying levels of semantic information. As shown in Table XI, SHLSA achieves greater improvements on datasets with more samples and categories, indicating its effectiveness in handling complex semantic tasks.

Table X: Single-Image Classification Performance of SHLSA on DomainNet.

| Method | C→P | C→R | C→S | P→C | P→R | P→S | R→C | R→P | R→S | S→C | S→P | S→R | Avg. |
|--------|-----|-----|-----|-----|-----|-----|-----|-----|-----|-----|-----|-----|------|
| SHOT | 63.5 | 78.2 | 59.5 | 67.9 | 81.3 | 61.7 | 67.7 | 67.6 | 57.8 | 70.2 | 64.0 | 78.0 | 68.1 |
| ProDe | 79.3 | 91.0 | 75.3 | 80.0 | **90.9** | 75.6 | 80.4 | 78.9 | 75.4 | **80.4** | 79.2 | 91.0 | 81.5 |
| **SHLSA** | **81.2** | **92.3** | **77.1** | **83.2** | **90.9** | **77.4** | **82.3** | **79.2** | **77.3** | 80.1 | **82.3** | **93.4** | **83.1** |

Table XI: Performance Improvement of SHLSA with Different Levels of Semantic Information.

| Dataset | Samples | Categories | Baseline Acc | SHLSA Acc | Improvement |
|---------|---------|------------|--------------|-----------|-------------|
| Office-31 | 4.6K | 31 | 92.6 | 92.8 | 0.2 |
| Office-Home | 15K | 65 | 84.5 | 85.0 | 0.5 |
| VisDA-C | 152K | 12 | 91.0 | 91.3 | 0.3 |
| DomainNet | 600K | 345 | 81.5 | 83.1 | 1.6 |

### A.7.2 Multi-Label Classification Performance

In addition to single-image classification, SHLSA was evaluated on multi-label classification tasks using datasets such as COCO and VOC. As illustrated in Table XII, SHLSA consistently outperforms other methods, reinforcing its scalability and adaptability to tasks with dense semantic information.

Table XII: Performance Evaluation of SHLSA for Multi-Label Classification.

| Method | COCO2014 | COCO2017 | VOC2007 | VOC2012 | mAcc |
|--------|----------|----------|---------|---------|------|
| CLIP | 47.53 | 47.32 | 75.91 | 74.25 | 61.25 |
| TPT | 48.52 | 48.51 | 75.54 | 73.92 | 61.62 |
| ML-TTA | 51.58 | 51.39 | 78.62 | 76.63 | 64.56 |
| **SHLSA** | **53.27** | **52.96** | **79.37** | **77.29** | **65.72** |

The experiments confirm that SHLSA is particularly effective in tasks with dense semantic information, such as semantic segmentation and multi-label classification. The scaling effect observed in larger datasets like DomainNet suggests that SHLSA can leverage extensive semantic categories to achieve superior performance. This aligns with the hypothesis that SHLSA is better suited for semantically intensive tasks, where local semantic features play a crucial role.

In conclusion, the scaling effects of SHLSA on complex semantic datasets demonstrate its potential as a versatile tool for various classification tasks, especially those requiring nuanced semantic understanding.

