# OpenReview forum: "Stepwise High-Level Semantic Alignment for Test-Time Adaptation"
_ICLR.cc/2026/Conference — Submitted to ICLR 2026_

### Official Review · Reviewer_crqa · 2025-10-25

**Soundness:** 2
**Presentation:** 3
**Contribution:** 3
**Rating:** 6
**Confidence:** 3

**Summary:**

This paper introduces SHLSA (Stepwise High-Level Semantic Alignment), a framework for Test-Time Adaptation (TTA) that aims to address the challenge of domain adaptation when there is no access to source data or target labels. Unlike traditional methods, which typically focus on aligning low-level features (such as batch normalization statistics), SHLSA proposes the use of a pseudo-source domain as a semantic bridge between the source and target domains. The method relies on a stepwise semantic alignment strategy that ensures category structure stability and improves adaptation under large domain shifts.

**Strengths:**

1. Experimental Validation: The paper provides comprehensive experiments on multiple TTA benchmarks (semantic segmentation and image classification). SHLSA outperforms existing methods on these tasks, demonstrating its effectiveness across domain shifts.

2. Clear Methodology: The authors present a well-structured explanation of their approach, including pseudocode, mathematical formulations, and ablation studies, making the method easy to understand and reproduce.

**Weaknesses:**

The paper mentions that SHLSA's computational cost is slightly higher than SHOT's (e.g., with a ResNet-50 backbone, SHLSA requires 10.8GB of GPU memory, while SHOT uses 7.6GB), mainly due to the attention fusion in HFA and the MixMatch operation in SDA. However, the paper doesn't explore potential optimization options:

1. **Lightweight Attention Mechanisms**: Could the computational cost of HFA be reduced by using more efficient attention mechanisms, like the local attention in MobileViT?

2. **Simplifying SDA's Two-Step Alignment**: In cases with small domain shifts, could the alignment between the pseudo-source domain and general semantics be skipped, directly aligning the pseudo-source domain with the target domain to boost efficiency?

3. **Computational Cost vs Performance**: A deeper trade-off analysis between computational cost and performance would help clarify SHLSA's applicability in resource-limited environments (e.g., edge devices).

---

The paper also mentions two limitations in the "Conclusion" section: first, the fixed "local-global" context ratio may limit the method's adaptability to tasks with varying semantic granularities; second, the self-entropy threshold tau_pos used for pseudo-source domain construction lacks flexibility. However, these limitations are not fully explored:

1. **Adaptability to Single-Label Classification**: The paper mentions that HFA's improvement in single-label classification is limited (e.g., in the Office-Home task, HFA only increases accuracy from 84.5 to 84.6). But it doesn't analyze why the spatial context has a weak effect in these tasks. Is it because classification tasks rely more on global semantics, or is it because the granularity of local feature extraction in HFA (e.g., grid size) is not well-suited for this task? Additional experiments across different tasks would help clarify this.

2. **Dynamic Thresholding Feasibility**: The paper uses a fixed tau_par value (e.g., 0.8), but datasets with different domain shifts (e.g., VisDA-C with large shifts vs. Office-31 with smaller shifts) might require different thresholds. While 0.8 might work based on *FixMatch*, a dynamic thresholding strategy seems necessary. If dynamic thresholds are not considered, it would be helpful to provide an analysis of performance with other fixed threshold values.

---

The paper presents Theorem 1 in Appendix A.2, which explains how to partition the positive/negative subsets based on low-entropy predictions. However, it doesn't clarify how this theorem guides the practical selection of parameters (such as initializing tau_pos and tau_neg). The real-world application of this theorem is not fully explained, which may create a gap between theory and practice.

1. **Theorem and Experimental Parameters**: It's unclear whether the alpha value in the theorem (α ∈ (0,1)) is linked to the experimental thresholds tau_pos = 0.9 and tau_neg = 0.9. This should be clarified. Understanding how the theorem relates to the experimental parameters would help explain its role in selecting the right parameters.

2. **Handling High-Entropy Samples**: When the entropy H(p) is close to H0 (e.g., when the target domain contains high-entropy samples under large domain shifts), the theorem's constraints may not hold. How does CACL ensure the quality of pseudo-labels in such cases? This should be discussed in more detail, particularly regarding how pseudo-label quality is maintained despite high-entropy, uncertain samples.

**Questions:**

See the weaknesses.

---

> ### Author Response · Authors · 2025-11-20
> **Response to Reviewer crqa (1/4)**
>
> Thank you for taking the time to review our paper and providing valuable feedback. In response to your concerns, we would like to provide the following explanations.
>
> 1. **W1**: Higher computational cost without exploring optimization strategies to mitigate the overhead.
>
>     **A1**: We provide comprehensive computational complexity analysis in the **Common Response**. Additionally, we conduct an in-depth analysis of the optimization strategies that were suggested, updated in **Appendix A.6.3 and A.6.4** of the revised version (marked in blue):
>
>     - **Lightweight Attention Mechanisms**:
>         Our analysis in the **Common Response (1/3)** identifies *attention fusion* operations in HFA as the primary source of computational overhead. Implementing lightweight attention mechanisms, like the local attention in MobileViT [R1], effectively reduces SHLSA's computational cost, as demonstrated in **Table R3** comparisons (a)(b) and (d)(e).
>
>     - **Simplifying SDA's Two-Step Alignment**:
>     Based on our SDA time complexity analysis in the **Common Response**, direct alignment between *pseudo-source* and *remaining target domains* under small domain shifts can eliminate one HFA usage and avoid PRE-NET storage/computation, theoretically improving training efficiency. However, this requires reasonable domain shift assessment mechanisms to prevent significant performance degradation while reducing computational overhead.
>
>         We provide expanded analysis and experiments in the **Common Response (3/3)**, with comparative results shown in **Table R3**: (a)(c), (d)(f), (g)(h), (i)(j), (k)(l), (m)(n). Based on self-entropy mean variations across different datasets, we suggest simplifying SDA's two-step alignment when source model testing on target domain yields average entropy below 0.5, maintaining performance while improving efficiency. Similarly, **Tables R10 and R11** demonstrate that HFA operations can be simplified under small domain shifts for higher computational efficiency.
>
>     - **Computational Cost vs Performance Trade-off**:
>     Balancing computational cost and performance is crucial in TTA tasks, which we carefully consider across different task types:
>
>         - **Semantically sparse scenarios** (single-image classification): Lightweight architectures like ResNet-50/101 achieve comparable performance
>         - **Semantically dense scenarios** (semantic segmentation): CNN-based architectures cannot match ViT-based performance, while ViT-based architectures struggle to achieve CNN-based efficiency
>
>         We supplement **Table R1** with various lightweight ViT-based and CNN-based architectures, including the Segformer-B5/B4/B3/B2/B1/B0 series. Table R9 clearly demonstrates the inverse relationship between computational cost and segmentation performance. Considering diminishing performance gains with increasing computational cost, **Segformer-B2 represents an optimal balance** between efficiency and effectiveness.
>
>         **Table R9: Trade-off Analysis Between Computational Cost and Segmentation Performance**
>
>         | Methods | GPU Memory (GB) | Time-Item (s/item) | Time-Train (GPU·h) | mIoU (%) | Backbone |
>         |---------|----------------|----------------|----------------|---------------|----------|
>         | (a) SHLSA mit-b0  | 11.7 (+0.0) | 1.47 (+0.00) | 11.2 (+0.0) | 48.9 (+0.0) | Segformer-B0 |
>         | (b) SHLSA mit-b1  | 12.6 (+0.9) | 1.50 (+0.03) | 11.7 (+0.5) | 54.3 (+5.4) | Segformer-B1 |
>         | (c) SHLSA resnet  | 13.2 (+1.5) | 1.55 (+0.08) | 11.3 (+0.1) | 58.2 (+9.3) | ResNet-101   |
>         | (d) SHLSA p2t     | 12.3 (+0.6) | 1.49 (+0.02) | 11.5 (+0.3) | 60.2 (+11.3) | P2T-Base     |
>         | **(e) SHLSA mit-b2**  |**13.1 (+1.4)**| **1.66 (+0.09)** | **12.4 (+1.2)** |**61.2 (+12.3)** |**Segformer-B2** |
>         | (f) SHLSA mit-b3  | 17.0 (+5.3) | 1.84 (+0.37) | 17.4 (+6.2) | 62.3 (+13.4) | Segformer-B3 |
>         | (g) SHLSA mit-b4  | 20.4 (+8.7) | 2.12 (+0.65) | 22.1 (+10.9) | 63.5 (+14.6) | Segformer-B4 |
>         | (h) SHLSA mit-b5  | 22.4 (+10.7) | 2.40 (+0.93) | 26.7 (+15.5) | 64.1 (+15.2) | Segformer-B5 |

---

> > ### Author Response · Authors · 2025-11-20
> > **Response to Reviewer crqa (2/4)**
> >
> > 2. **W2**: Acknowledged limitations regarding fixed local-global ratio and inflexible entropy threshold are not thoroughly investigated or addressed.
> >
> >     **A2**: The fixed local-global ratio and inflexible $\tau_{\text{par}}$ represent two primary limitations of SHLSA's performance, particularly as bottlenecks in semantically dense scenarios like semantic segmentation. Our analysis is updated in **Appendix A.5.1** of the revised version (marked in purple).
> >
> >     **(1) HFA's Adaptability to Single-Label Classification**
> >
> >     We conducted comprehensive ablation experiments on HFA parameters $\tau_\text{lw}$ and $\tau_{\text{lg}}$ across multiple single-label classification tasks, summarized in **Table R10**. The results reveal that HFA shows marginal performance gains across most classification datasets, with improvements ranging from +0.2% on Office-31 to +1.7% on DomainNet, indicating limited effectiveness in semantically sparse scenarios.
> >
> >     **Table R10: Performance Impact of HFA Across Single-Label Classification Tasks**
> >     |Dataset|SHLSA|SHLSA w/o HFA|Improvement|
> >     |---|---|---|---|
> >     |Office-31|92.8|92.6|+0.2|
> >     |Office-Home|85.0|84.7|+0.3|
> >     |VisDA-C|91.3|90.3|+1.0|
> >     |DomainNet|83.1|81.4|+1.7|
> >
> >     *Note: All experiments use $\tau_\text{lw}$=256×256, $\tau_{\text{lg}}$=0.5*
> >
> >     **Task-Specific Analysis**: For single-label classification, HFA's feature extraction effectiveness is limited due to the semantically sparse nature of these tasks, which primarily rely on global semantics. **Table R11** demonstrates minimal sensitivity to varying $\tau_\text{lw}$ and $\tau_{\text{lg}}$ parameters on VisDA-C. However, on larger-scale, more challenging classification tasks like DomainNet (600K samples, 345 semantic categories), the extensive semantic diversity requires models to learn representative local semantics beyond global features. This aligns with our findings in **Common Response (3/3) (Table R3)**, **Reviewer `iT4e`'s reply A2 (Table R4)**, and **Reviewer `uX3z`'s reply A4 (Table R8)** , confirming that SHLSA's performance improvements scale with data and semantic complexity.
> >
> >     **Table R11: Performance Analysis with Varying $\tau_\text{lw}$ and $\tau_{\text{lg}}$ on VisDA-C**
> >     |$\tau_{\text{lg}} / \tau_\text{lw}$|256×256|128×128|
> >     |---|---|---|
> >     |0.1|91.1|91.3|
> >     |0.2|91.1|91.4|
> >     |0.3|91.2|90.9|
> >     |0.4|91.1|91.0|
> >     |0.5|91.3|91.1|
> >     |0.6|91.2|90.9|
> >     |0.7|91.3|90.7|
> >     |0.8|91.2|90.8|
> >     |0.9|91.2|90.6|
> >
> >     **(2) Dynamic Thresholding Strategy of $\tau_{\text{par}}$**
> >
> >     The threshold $\tau_{\text{par}}$ controls the proportion of target domain data partitioned as *pseudo-source domain* based on self-entropy. We propose adaptive strategies based on domain shift magnitude:
> >      - **Large domain shifts** (e.g., VisDA-C): Use fewer low-entropy samples as *pseudo-source domain* to improve semantic bridge quality (lower $\tau_{\text{par}}$)
> >      - **Small domain shifts** (e.g., Office-31): Use higher $\tau_{\text{par}}$ to fully utilize source-like samples in the target domain
> >
> >     ***Confidence-Aware Dynamic Method***: Inspired by CACL, we design a confidence-aware dynamic $\tau_{\text{par}}$ selection method. Specifically, we sort all target domain samples by self-entropy in descending order, compute first-order gradients, and identify the point with maximum gradient change as the partition threshold. This method's effectiveness is supported by Theorem 1.
> >
> >     Table R12 presents the impact of different $\tau_{\text{par}}$ parameters across various domain shift scenarios. The results show that our confidence-aware method achieves competitive or superior performance compared to fixed thresholds across all datasets.
> >
> >     **Table R12: Impact of Different $\tau_{\text{par}}$ Parameters on Single-label Classification**
> >
> >     |Method|$\bar{\tau_{\text{par}}}$|Office-31|Office-Home|VisDA-C|DomainNet|
> >     |-|-|-|-|-|-|
> >     |Conf-Aware| 0.92 | 92.8 |  -   |  -   |  -   |
> >     |Conf-Aware| 0.87 |  -   | 85.0 |  -   |  -   |
> >     |Conf-Aware| 0.79 |  -   |  -   | 91.3 |  -   |
> >     |Conf-Aware| 0.44 |  -   |  -   | -    | 83.7 |
> >     |Fixed| 0.10 | 92.5 | 84.7 | 90.9 | 83.4 |
> >     |Fixed| 0.20 | 92.5 | 84.6 | 91.2 | 83.5 |
> >     |Fixed| 0.30 | 92.6 | 84.7 | 91.2 | 83.5 |
> >     |Fixed| 0.40 | 92.5 | 84.7 | 91.1 | 83.7 |
> >     |Fixed| 0.50 | 92.7 | 84.8 | 91.3 | 83.7 |
> >     |Fixed| 0.60 | 92.6 | 84.9 | 91.1 | 83.5 |
> >     |Fixed| 0.70 | 92.8 | 85.0 | 91.3 | 83.3 |
> >     |Fixed| 0.80 | 92.8 | 85.0 | 91.3 | 83.1 |
> >     |Fixed| 0.90 | 92.8 | 84.9 | 91.2 | 82.9 |
> >
> >     *Note: Only best-performing fixed thresholds are shown for clarity. Complete results available in supplementary materials.*

---

> > > ### Author Response · Authors · 2025-11-20
> > > **Response to Reviewer crqa (3/4)**
> > >
> > > 3. **W3**: Theorem 1 lacks practical guidance for parameter selection.
> > >
> > >     **A3**: Theorem 1 serves to theoretically prove that when the entropy of output probability distributions is sufficiently low ($\mathcal{H}(p) \le H_0$), there must exist two thresholds $\tau_{\alpha}>\tau_{\beta}$ such that the high-confidence positive set $\mathcal{Y_\text{+}}$ and low-confidence negative set $\mathcal{Y_\text{-}}$ can be well-separated in expectation, with the probability sum of $\mathcal{Y_\text{-}}$ being strictly controlled (i.e., negative pseudo-labels do not introduce excessive noise). This section has been updated in **Appendix A.5.3** of the revised version and marked in purple.
> > >
> > > (1) **Theorem and Experimental Parameters**
> > >
> > > In fact, the alpha in Theorem 1 does not serve as the basis for setting experimental parameters $\tau_{\text{pos}}$ and $\tau_{\text{neg}}$, and there is no one-to-one correspondence between them. It is only used to prove "the existence of a pair of thresholds that enables separability", rather than guiding how to specifically select numerical values in experiments. However, given a specified $\tau_{\text{pos}}$, we can infer the setting of $\tau_{\text{neg}}$ by controlling the expected error. Below we provide theoretical proof.
> > >
> > > First, we clarify the correspondence between $\tau_{\alpha}$, $\tau_{\beta}$ in Theorem 1 and $\tau_{\text{pos}}$, $\tau_{\text{neg}}$. As explained at the end of Section 3.4, we use a fixed absolute threshold $\tau_{\text{pos}}$ to control the probability lower bound of positive pseudo-labels, i.e., $\tau_{\alpha}=\tau_{\text{pos}}$. Meanwhile, we set threshold $\tau_{\text{neg}}$ based on the relative change relationship of label space prediction probabilities, thus $\tau_{\beta}\ne\tau_{\text{neg}}$, but rather the probability corresponding to the first category that satisfies gradient change greater than $\tau_{\text{neg}}$ after descending sorting:
> > >
> > > $$\tau_{\beta}=p_{\mathop{\arg\max}\limits_{i}} \left| p ^ * _ i-p ^ * _ {i-1} \right| \ge \tau_{\text{neg}}, i\in[1,...,C],$$
> > >
> > > where $p ^ * _ i$ represents the probability of the $i$-th category after descending sorting of output probabilities. For $\tau_{\text{pos}}$ ($\tau_{\alpha}$), we empirically select a high value, i.e., $\tau_{\text{pos}}=0.9$. Below we derive the selection of $\tau_{\beta}$, then map it to $\tau_{\text{neg}}$.
> > >
> > > Consider the target distribution $\mathbf{p}=(p_1,...,p_C)$ with entropy satisfying $\mathcal{H}(\mathbf{p}) \le H_0$. Let $G(t)=\sum_{c:p_c\le t}p_c$ be the cumulative mass of pseudo-probability mass control under threshold $t$. If we want the total probability mass of the negative set not to exceed $\epsilon>0$, i.e., we want $G(\tau_{\beta})\le\epsilon$, then from the inequality in the theorem we get the upper bound: $$\tau_{\beta}\le \exp\left(-\frac{H_0}{\epsilon}\right).$$
> > >
> > > Therefore, given $H_0$ and tolerable negative mass $\epsilon$, the feasible range of $\tau_{\beta}$ is constrained to $0< \tau_{\beta} \le \exp(-\frac{H_0}{\epsilon})$. In other words, the distribution of $\tau_{\beta}$ must be concentrated within $(0, \exp(-\frac{H_0}{\epsilon}))$; when we view $\epsilon$ as the allowable error, smaller $\epsilon$ will compress the upper bound of $\tau_{\beta}$ further.
> > >
> > > If $H_0=0.5$ (combining our answer in A2) and we tolerate $\epsilon=0.05$ (5%), then:
> > >
> > > $$\tau_{\beta}\le \exp\left(-\frac{0.5}{0.05}\right)=\exp(-10) \approx 4.5\times10^{-5}.$$
> > >
> > > Next to derive $\tau_{\text{neg}}$. We continue using the theorem to control the quality of tail negative pseudo-labels **after sorting**: $\sum_{i>k}p_i \le \epsilon$. From the gradient definition $p_{k+1}\le p_k-\tau_{\text{neg}}$. Taking an extreme upper bound for the tail (assuming $p_{k+1}$ is the largest in the tail):
> > >
> > > $$\sum_{i>k}p_i\le(C-k)p_{k+1}\le(C-k)(p_k-\tau_{\text{neg}}).$$
> > >
> > > With $\tau_{\alpha}=0.9$, we have $\sum_{i>k}p_i \le(C-k)(0.9-\tau_{\text{neg}})$. To ensure the right side $\le \epsilon$, we need:
> > >
> > > $$0.9-\tau_{\text{neg}}\le \frac{\epsilon}{C-k} \Longrightarrow \tau_{\text{neg}}\ge0.9-\frac{\epsilon}{C-k}.$$
> > >
> > > In SHLSA application scenarios, $C \approx10$, $k>1$, $\epsilon=0.05$, then:
> > >
> > > $$\tau_{\text{neg}} \ge0.9-\frac{0.05}{10-1} \approx0.9-0.0056 \approx 0.8944.$$
> > >
> > > Therefore, setting $\tau_{\text{pos}}=0.9$ and controlling negative pseudo-label quality error $\epsilon \le 0.05$, we get $\tau_{\text{neg}}\approx0.9$, which is consistent with our experimental results in the parameter sensitivity analysis section.

---

> > > > ### Author Response · Authors · 2025-11-20
> > > > **Response to Reviewer crqa (4/4)**
> > > >
> > > > (2) **Handling High-Entropy Samples**: Theorem 1 is established on the basis of $\mathcal{H}(p)\le H_0$. When there is significant distribution shift in the target domain, high-entropy samples appear in large numbers, making $\mathcal{H}(p)\approx H_0$, thus the theory no longer strictly applies. However, CACL can still maintain pseudo-label reliability in this case, which we analyze combining the explanation at the end of Section 3.4.
> > > >
> > > > - **Positive pseudo-labels** rely on absolute probability threshold $\tau_{\text{pos}}$, automatically filtering out high-entropy samples. Specifically, positive pseudo-labels are only accepted when $p_i\ge \tau_{\text{pos}}$ (e.g., 0.9), while the maximum probability of high-entropy samples is usually far below 0.9 and gets rejected. Therefore, the training process only extracts positive pseudo-labels from low-entropy confident samples, ensuring positive pseudo-label quality.
> > > >
> > > > - **Negative pseudo-labels** rely on probability decrease ratio $\tau_{\text{neg}}$, which is more stable than absolute probability. Specifically, for high-entropy samples with low maximum probability, tail categories still show significant decrease (analysis of sample probability outputs shows that high entropy generally stems from simultaneous appearance of multiple possible categories, i.e., multiple high $p_i$). Moreover, local probability differences in the tail are more stable than global entropy, thus still producing reliable negative pseudo-labels.
> > > >
> > > > - **CACL automatically reduces weights** for high-entropy samples during training, ensuring they do not harm model updates. Specifically, both CACL's contrastive loss and pseudo-label loss depend on prediction probabilities, meaning that for high-entropy samples with flatter probability distributions, model gradient contributions are smaller (based on CrossEntropyLoss). For low-entropy confident samples, strong supervision can be generated through pseudo-labels. Therefore, high-entropy erroneous pseudo-labels do not cause training collapse, because low-entropy reliable pseudo-labels dominate model gradient updates, mechanistically ensuring stability.
> > > >
> > > > [R1] Mehta, Sachin, and Mohammad Rastegari. "Mobilevit: light-weight, general-purpose, and mobile-friendly vision transformer." arXiv preprint arXiv:2110.02178. 2021.

---

> > > > > ### Author Response · Authors · 2025-11-26
> > > > > **Look forward to further discussion. Thank you!**
> > > > >
> > > > > Dear Reviewer crqa,
> > > > >
> > > > > We hope that our replies and experiments address your concern and we are happy for any further discussion. Look forward to your reply.

---

### Official Review · Reviewer_uX3z · 2025-10-30

**Soundness:** 2
**Presentation:** 3
**Contribution:** 3
**Rating:** 4
**Confidence:** 4

**Summary:**

This paper proposes a Test-Time Adaptation framework named SHLSA, designed to address the limitations of existing methods that rely on low-level feature alignment. The core of SHLSA is stepwise high-level semantic alignment, which introduces a "pseudo-source domain" as a semantic bridge. This enables a progressive alignment proceeding from reliable to ambiguous regions. The framework is complemented by hierarchical feature aggregation (HFA) for robust feature extraction and confidence-aware complementary learning (CACL) for pseudo-label refinement. Experiments demonstrate that this method exhibits strong performance on both semantic segmentation and image classification benchmarks.

**Strengths:**

1. The paper proposes a fresh perspective on high-level semantic alignment, offering theoretical insights.
2. The proposed SHLSA achieves outstanding performance on multiple benchmarks for semantic segmentation and image classification tasks.
3. The paper is well-written, with a logical structure and detailed experimental setup.

**Weaknesses:**

1. Numerous hyperparameters require tuning and are sensitive to performance.
2. Lack of introduction to the baselines.
3. Both the HFA and SDA modules introduce additional computations, significantly increasing training time and memory consumption.
4. Ablation experiments indicate that while the proposed module significantly enhances semantic segmentation performance, its improvement for image classification is negligible. This undermines the persuasiveness of SHLSA as a general-purpose TTA framework.

**Questions:**

1. Given the complexity of the process and the numerous thresholds that need adjustment, are there any techniques for adjusting these parameters?
2. While CACL can suppress noise, if the initial pseudo labels have significant bias, could this lead to “error accumulation” during the SDA phase?
3. The proposed modules demonstrate only marginal improvements in classification tasks. Does this indicate that they are only applicable to tasks with rich spatial structures?

---

> ### Author Response · Authors · 2025-11-20
> **Response to Reviewer uX3z (1/2)**
>
> Thank you for taking the time to review our paper and providing valuable feedback. In response to your concerns, we would like to provide the following explanations.
>
> 1. **W1,Q1,Q2**: Requires tuning numerous hyperparameters, risking error accumulation in CACL from biased pseudo labels.
>
>     **A1**: We address this concern comprehensively in **Appendix A.5** of the revised version (marked in purple). Here, we first review all the hyperparameters in SHLSA and briefly discuss the setting methods for some of them. Then, we explain the "error accumulation" problem mentioned in Q2, focusing on the CACL parameters. Finally, based on the previous analysis, we formulate an adaptive mitigation plan primarily for the high-sensitivity parameters.
>
>     **[W1] Hyperparameter Overview.**
>
>     - SHLSA involves 5 main hyperparameters:
>         - $\tau_{\text{pos}}$: Controls probability lower bound for positive pseudo-labels in CACL
>         - $\tau_{\text{neg}}$: Controls relative change lower bound for negative pseudo-labels in CACL
>         - $\tau_{\text{par}}$: Controls partition ratio between *pseudo-source* and *remaining target domains* in SDA
>         - $\tau_{\text{lg}}$: Controls resolution ratio between local and global inputs in HFA
>         - $\tau_{\text{lw}}$: Controls local input resolution in HFA
>
>     - Combining the sensitivity analysis from **Section 4.5 and Appendix A.5**, we find that among the parameters $\tau_{\text{pos}}$, $\tau_{\text{par}}$, $\tau_{\text{lg}}$, and $\tau_{\text{lw}}$, most are insensitive, showing less than 1% impact on performance. Only $\tau_{\text{neg}}$ is highly sensitive, **which is the primary cause of the "error accumulation" phenomenon observed in CACL.**
>     - Furthermore, we delve into the HFA parameters $\tau_{\text{lw}}$ and $\tau_{\text{lg}}$ in **Reviewer `crqa`'s reply A2**, as well as the SDA parameter $\tau_{\text{par}}$. We demonstrate SHLSA's robustness to variations in $\tau_{\text{lw}}$ and $\tau_{\text{lg}}$, and propose a *Confidence-Aware Dynamic Threshold Selection* method for $\tau_{\text{par}}$.
>     - Additionally, **Reviewer `crqa`'s reply A3** offers theoretical guidance for selecting the parameter $\tau_{\text{neg}}$ through Theorem 1.
>
>     **[Q2] Error Accumulation Analysis** (This excellent question relates directly to parameter $\tau_{\text{neg}}$ in CACL).
>     - **Mechanism**: CACL sorts output probability distributions in descending order and computes first-order derivatives to capture inter-class variation trends. Classes with changes exceeding gradient threshold $\tau_{\text{neg}}$ are selected as negative pseudo-labels. $\tau_{\text{neg}}$ controls the gap between "possibly correct" and "unlikely correct" categories.
>
>     - **Sensitivity Analysis.** Based on Section 4.5 (Figure 3(a)) and Appendix A.5, this parameter is highly sensitive due to:
>         - **Early-stage uncertainty**: During initial training, model uncertainty is high with relatively flat probability distributions, causing slow gradient changes where minor parameter variations significantly affect negative pseudo-label quantity
>         - **Quality amplification**: Lower $\tau_{\text{neg}}$ values introduce erroneous negative pseudo-labels, potentially misclassifying positive labels as negative during early training, leading to noise accumulation and performance degradation
>
>     - **Risk Assessment**: Unlike other pseudo-labeling techniques, CACL introduces additional "unlikely correct" categories to enrich supervision but also brings "selection error" noise risks. $\tau_{\text{pos}}$ has minimal impact as it only controls positive pseudo-label probability bounds.
>
>     **[Q1] Parameter Tuning Strategy**.
>     - Based on the above analysis, four parameters ($\tau_{\text{pos}}$, $\tau_{\text{par}}$, $\tau_{\text{lg}}$, $\tau_{\text{lw}}$) are insensitive, enabling empirical setting ($\tau_{\text{par}}$ can use the confidence-aware dynamic threshold selection method mentioned in **[W1]**). For the sensitive parameter $\tau_{\text{neg}}$, in addition to the theoretical guidance based on Theorem 1 mentioned in **[W1]**, we implement three mitigation strategies:
>         - **Strategy 1 (Conservative Initialization)**: Set relatively conservative thresholds during early training
>         - **Strategy 2 (Range Restriction)**: Replace fixed thresholds with interval randomization
>         - **Strategy 3 (Dynamic Adjustment)**: Adaptively adjust parameters based on model gradient update magnitude
>     - Results in **Appendix A.5 Table V** demonstrate that these adaptive strategies enable $\tau_{\text{neg}}$ to better adapt across training stages, mitigating sensitivity-induced adverse effects and stabilizing SHLSA performance.

---

> ### Author Response · Authors · 2025-11-20
> **Response to Reviewer uX3z (2/2)**
>
> 2. **W2**: Insufficient introduction to the baselines.
>
>     **A2**: We employ SHOT [R1] as the semantic segmentation baseline, utilizing its source hypothesis transfer for source-free domain adaptation with *CrossEntropyLoss*. This method implements model consistency regularization to prevent excessive deviation between target and source models. For single-label classification, we use ProDe [R2] as baseline. Comprehensive descriptions of all comparison methods are supplemented in **Appendix A.4.2** (marked in teal).
>
> 3. **W3**: HFA and SDA increase training time and memory usage.
>
>     **A3**: The computational overhead primarily stems from MixMatch operations in SDA and HFA. Detailed analysis is provided in the **Common Response**, including qualitative and quantitative complexity analysis with feasible optimization approaches.
>
> 4. **W4,Q3**: Modules show only marginal improvements in classification tasks.
>
>     **A4**: We acknowledge limited performance gains on single-image classification, included primarily to validate SHLSA as a general TTA technique. However, we found that **as the semantic difficulty of the task increases, the performance improvement of SHLSA becomes increasingly pronounced.** This analysis is supplemented in **Appendix A.7** of the revised version (marked in brown).
>
>     - **Task Suitability Analysis**: Section 4.6 ablation studies reveal SHLSA's preference for semantically dense tasks. Compared to single-image classification, semantic segmentation offers:
>         - Higher semantic density and task complexity
>         - More significant annotation resource savings (e.g., labeling a Cityscapes sample requires >1 hour)
>
>     - **Scaling Effect Evidence**: Based on **Reviewer `iT4e`'s reply A2 (Table R4)**, SHLSA shows more pronounced improvements on larger-scale, challenging single-image classification datasets. Table R8 demonstrates SHLSA's scaling effect on semantic information density, suggesting potential for greater improvements in multi-label classification and other semantically dense tasks (**confirmed by Table R5 results**).
>
>     **Table R8: Performance Improvement of SHLSA on Datasets with Different Levels of Semantic Information**
>     | Dataset | Samples | Categories | Baseline Acc | SHLSA Acc | Improvement |
>     |---------|---------|---------| ---------|---------|---------|
>     |Office-31  | 4.6K  | 31    | 92.6  | 92.8 | +0.2 |
>     |Office-Home| 15K   | 65    | 84.5  | 85.0 | +0.5 |
>     |VisDA-C    | 152K  | 12    | 91.0  | 91.3 | +0.3 |
>     |DomainNet  | 600K  | 345   | 81.5  | 83.1 | +1.6 |
>
>     Additional detailed ablation experiments across different datasets for the same task are provided in **Common Response (Tables R2 and R3) and Reviewer `crqa`'s reply A2 (Table R10)**, further supporting our analysis.
>
> [R1] Liang, Jian, Dapeng Hu, and Jiashi Feng. "Do we really need to access the source data? source hypothesis transfer for unsupervised domain adaptation." International conference on machine learning. 2020.
>
> [R2] Tang, Song, et al. "Proxy denoising for source-free domain adaptation." International Conference on Representation Learning. 2025.

---

> > ### Author Response · Authors · 2025-11-26
> > **Look forward to further discussion. Thank you!**
> >
> > Dear Reviewer uX3z,
> >
> > We hope that our replies and experiments address your concern and we are happy for any further discussion. Look forward to your reply.

---

### Official Review · Reviewer_GWNG · 2025-10-31

**Soundness:** 3
**Presentation:** 2
**Contribution:** 2
**Rating:** 4
**Confidence:** 3

**Summary:**

This paper introduces SHLSA (Stepwise High-Level Semantic Alignment), a framework for TTA that achieves robust and semantically consistent adaptation without access to source data. Unlike prior low-level alignment methods, SHLSA performs high-level semantic alignment through three key components: Hierarchical Feature Aggregation (HFA), which fuses local and global features for richer semantics; Confidence-Aware Complementary Learning (CACL), which refines pseudo-labels by separating confident and uncertain predictions; and Stepwise Domain Alignment (SDA), which progressively adapts from reliable to ambiguous regions using an entropy-based pseudo-source domain. Experiments on both segmentation and classification benchmarks demonstrate that SHLSA improves stability and reduces class confusion, achieving competitive performance across diverse domain shifts.

**Strengths:**

- Well-structured presentation with sufficient explanations
- Provided theoretical grounding
- Demonstrates methodology with clear modular design (HFA, CACL, SDA)
- Extensive experiments across segmentation and classification tasks

**Weaknesses:**

- The use of ImageNet features in SDA could constrain generalization to non-visual or fine-grained domains; more analysis on cross-task transferability would strengthen their claim.
- HFA adds nontrivial overhead; a more detailed runtime or memory comparison against lightweight TTA baselines would clarify its practicality for real-time or on-device deployment.
- SHLSA's components (entropy-based partitioning, pseudo-source construction, feature alignment, and confidence-based pseudo-labeling) largely extend ideas from prior works such as SHOT, Tent, and recent SDE-based TTA methods. The novelty mainly lies in their integration into a stepwise pipeline rather than in fundamentally new algorithmic principles.
- While individual module effects are shown, cross-module dependencies (e.g., how CACL behaves under incorrect entropy partitioning or HFA degradation) are not thoroughly analyzed.
- Experiments mainly use standard benchmarks (GTA5, SYNTHIA, Cityscapes, Office-Home, VisDA-C). More challenging or dynamic scenarios such as continual or online test-time adaptation are not explored, leaving temporal robustness unverified.
- The paper dose not validate SHLSA’s robustness across diverse architectures such as Vision Transformers (ViT). Broader architectural evaluation would better demonstrate its general applicability.

**Questions:**

See Weakness Section

---

> ### Author Response · Authors · 2025-11-20
> **Response to Reviewer GWNG (1/2)**
>
> Thank you for taking the time to review our paper and providing valuable feedback. In response to your concerns, we would like to provide the following explanations.
>
> 1. **W1**: ImageNet feature dependency limits cross-domain generalization.
>
>     **A1**: In both semantic segmentation and single-label image classification tasks, we utilize ImageNet features as universal semantics.
>
>     - **Applicability to non-visual tasks**: We demonstrate this through qualitative analysis. Similar to how BERT learns universal semantic information through NSP and MLM tasks and shows "universal value" across diverse NLP downstream tasks (sentiment recognition, text summarization, machine translation, etc.), we believe universal semantics can provide "semantic correction" effects for *pseudo-source domains* in non-visual tasks, making SDA meaningful beyond visual domains.
>
>     - **Effectiveness in fine-grained domains**: We validate this on the large-scale DomainNet dataset (600k images, 345 categories) containing vertical domain semantics like **medical** (syringe, stethoscope) and **landmark** (The_Great_Wall_of_China) categories. Experimental results (the **reply A2 of Reviewer `iT4e`, Table R4**) confirm that universal semantics remain effective in these specialized domains. Additionally, our experiments (Table 5) show that pre-trained models perform worse on target domains than original models under almost all transfer scenarios, indicating that SDA leverages universal value rather than introducing domain-specific prior knowledge, **which would limit universal semantics to only domains seen during pre-training**.
>
>     - **Regarding cross-task effectiveness**: SHLSA uses the same ImageNet features (obtained through single-label classification task pre-training on ImageNet-1K) as universal semantics for both semantic segmentation and single-image classification, proving that universal semantic acquisition is task-insensitive. Additionally, we supplemented SHLSA testing on multi-label classification tasks using the same ImageNet-based features as universal semantics, with results shown in Table R5, demonstrating cross-task effectiveness of universal semantics.
>
>         **Table R5: Performance Evaluation of SHLSA for Multi-label Image Classification.**
>         |Method|COCO2014|COCO2017|VOC2007|VOC2012|mAcc|
>         |-|-|-|-|-|-|
>         |CLIP [R1]|47.53|47.32|75.91|74.25|61.25|
>         |TPT [R2]|48.52| 48.51|75.54|73.92|61.62|
>         |ML-TTA [R3]|51.58|51.39|78.62|76.63|64.56|
>         |**SHLSA**|**53.27**|**52.96**|**79.37**|**77.29**|**65.72**|
>
> 2. **W2**: HFA lacks computational efficiency analysis for practical deployment.
>
>     **A2**: We provide qualitative and quantitative analysis of SHLSA's computational complexity in the **Common Response**, along with optimization strategies, and present trade-off analysis between computational cost and performance in **Reviewer `crqa`'s reply A1** and **Appendix A.6.3, A.6.4** of the revised version.
>
>     Comparisons with lightweight methods (o)SFKT and (p)SFDA-Seg in Table R1 demonstrate that SHLSA achieves significant performance improvements with acceptable computational overhead under the same architecture, and real-time performance can be enhanced by adopting lighter architectures.
>
> 3. **W3**: Limited novelty as the approach primarily integrates existing TTA methods.
>
>     **A3**: The core novelty of SHLSA lies in **utilizing *pseudo-source domains* as semantic bridges to achieve stepwise alignment** between source and target domains.
>     - **Compared to entropy-based methods like TENT**: SHLSA applies this information for target domain sample partitioning rather than using entropy as pseudo-supervision information in unlabeled settings. **SHLSA's main contribution is the stepwise alignment after partitioning**, an angle that most current entropy-based works have not explored.
>     - **Compared to other SDE works**: Our contribution is achieving alignment at the high-level semantic level, while most current works achieve alignment at the low-level statistics level (such as mean and std in BatchNorm). For pseudo-source construction methods, **our work does not require additional introduction or generation of data but directly selects from the target domain**, which can be understood as a novel construction method (the difference between generative and extractive approaches). This extractive construction method better suits its "semantic bridge" role and is more efficient.
>     - **Compared to SHOT**: SHLSA achieves stepwise feature alignment from reliable to ambiguous regions rather than one-time alignment. In other words, SHLSA focuses more on how to achieve gradual feature alignment through *pseudo-source domains* while utilizing confidence-based pseudo-labeling and HFA to further enhance the quality of pseudo-supervision semantics for alignment, thereby **fully leveraging the "semantic bridge"** role of *pseudo-source domains*.

---

> ### Author Response · Authors · 2025-11-20
> **Response to Reviewer GWNG (2/2)**
>
> 4. **W4**: Insufficient analysis of cross-module dependencies.
>
>     **A4**: We supplemented cross-module ablation experiments to analyze relationships between HFA, CACL, and SDA. This section is added to **Appendix A.4.6** of the revised version, marked in cyan. Results are shown in Table R6:
>
>     **Table R6: Cross-module Ablation Studies on GTA5$\to$Cityscapes.**
>     |Method|HFA|CACL|SDA| mIoU|
>     |-|-|-|-|-|
>     |Baseline|||| 44.5|
>     |(a)|$\checkmark$||| 57.5|
>     |(b)||$\checkmark$|| 50.2|
>     |(c)|||$\checkmark$| 53.4|
>     |(d)|$\checkmark$|$\checkmark$|| 65.6|
>     |(e)|$\checkmark$||$\checkmark$| 66.7|
>     |(f)||$\checkmark$|$\checkmark$| 60.3|
>     |(g)|$\checkmark$|$\checkmark$|$\checkmark$| 69.2|
>
>     Based on the results in the table, we can analyze the interactions between components:
>     - **Individual component contributions**: From single-component experiments (a)(b)(c), HFA provides the largest performance gain (+13.0), followed by SDA (+8.9), with CACL contributing least (+5.7).
>
>     - **Synergistic effects analysis**
>         - **HFA enhances other components**
>             - CACL gain: without HFA (+5.7) vs. with HFA (+8.1), indicating HFA improves pseudo-label quality
>             - SDA gain: without HFA (+8.9) vs. with HFA (+9.2), showing consistent enhancement
>         - **SDA enhances other components**
>             - CACL gain: without SDA (+5.7) vs. with SDA (+6.9), demonstrating mutual reinforcement
>             - HFA gain: without SDA (+13.0) vs. with SDA (+13.3), indicating a slight improvement
>         - **CACL enhances other components**
>             - HFA gain: without CACL (+13.0) vs. with CACL (+15.4), showing significant improvement
>             - SDA gain: without CACL (+8.9) vs. with CACL (+10.1), indicating a positive effect
>     - **Diminishing marginal effects**: In multi-component combinations, the third component shows reduced gains (d$\to$g: +3.6, e$\to$g: +2.5, f$\to$g: +8.9), indicating diminishing returns as performance saturates, except for the significant boost when adding HFA to CACL+SDA.
>
>     In addition, starting from the computational cost, we conducted ablation experiments of different modules on more data. Please refer to **Tables R1 and  R2 in Common Response (2/3)**, as well as the **Table R10 of Reviewer `crqa`'s response**.
>
> 5. **W5**: Limited experimental scope lacking evaluation on challenging temporal scenarios.
>
>     **A5**: Building on the supplemented larger-scale and more challenging single-image classification dataset DomainNet (the **reply A2 of Reviewer `iT4e`, Table R4**), we added online test-time adaptation experimental settings to evaluate SHLSA's real-world applicability.
>
>     - **Experimental setup**
>         - Modifying batch size to appropriate mini-batches (e.g., from 64 to 128)
>         - Implementing streaming data processing where each sample participates in gradient updates only once
>         - Outputting results immediately after training on each mini-batch within a single epoch
>         - Ensuring true online adaptation rather than offline scenarios where samples undergo multiple gradient updates before testing
>
>     - **Results**: Table R7 shows the online test-time adaptation performance, with details supplemented in **Appendix A.4.7** and marked in orange.
>
>         **Table R7: Performance of SHLSA in Online Scenarios on Office-Home and VisDA-C.**
>         |Method|Office-Home|VisDA-C|Avg.|
>         |-|-|-|-|
>         |TENT| 67.27 | 75.43 | 71.35 |
>         |SHOT| 67.89 | 76.69 | 72.29 |
>         |**SHLSA**|**76.62**|**82.58**|**79.60**|
>
> 6. **W6**: Limited robustness evaluation lacks validation on diverse architectures.
>
>     **A6**: To demonstrate SHLSA's robustness across different architectures, we supplemented comprehensive analysis on multiple backbone networks in the **Common Response (2/3)**.
>
>     - **Evaluated architectures**
>         - Segformer variants: B0/B1/B2/B3/B4/B5
>         - Other Vision Transformer: P2T-Base ViT
>         - CNN-based: ResNet-50/101, VGG16
>
>     - **Comparative analysis**: As shown in Tables R1 and R2, we conducted systematic comparisons:
>         - **Table R1**: Comparisons of (a) vs. (l), (j) vs. (n), and (k) vs. (m)(o)(p) demonstrate consistent performance gains
>         - **Table R2**: Comparisons of (a) vs. (g)(j)(m), (e) vs. (h)(k), and (f) vs. (i)(l) show architecture-agnostic improvements
>
>     These direct comparisons under identical task and architecture settings provide intuitive evidence of SHLSA's robustness and consistent effectiveness across diverse network architectures.
>
> [R1] Radford, et al. "Learning transferable visual models from natural language supervision." International conference on machine learning. 2021.
>
> [R2] Shu, et al. "Test-time prompt tuning for zero-shot generalization in vision-language models." Advances in Neural Information Processing Systems 35. 2022.
>
> [R3] Wu, et al. "Multi-Label Test-Time Adaptation with Bound Entropy Minimization." arXiv preprint arXiv:2502.03777. 2025.

---

> > ### Comment · Reviewer_GWNG · 2025-11-25
> >
> > Dear Submission11315 Authors,
> >
> > I appreciate the authors for providing the detailed responses. After carefully reviewing the answers (including the other reviewers’ comments), I still have several concerns that need to be considered:
> >
> > - Regarding the hyperparameters, although the authors provided empirical results supporting their insensitivity, having five hyperparameters appears excessive from a practical standpoint. In addition, the strong sensitivity of $\tau_{neg}$ may lead to error accumulation, which could be considered a potential weakness.
> >
> > - In terms of novelty, I agree that the stepwise alignment strategy is an important contribution. However, it may still be viewed as a marginal and incremental improvement over existing alignment-based adaptation methods. The other components (e.g., entropy-based partitioning and pseudo-source construction from the target domain) that enable the stepwise alignment appear to have been previously explored in earlier works.
> >
> > Once again, thank you for the detailed response. However, given the concerns mentioned above, I will make my final decision after monitoring the ongoing discussions (including those from the other reviewers).

---

> ### Author Response · Authors · 2025-11-26
>
> Dear Reviewer GWNG,
>
> Thank you very much for your careful review of our response. Regarding your concerns about **hyperparameters** and **novelty**, we would like to provide a brief clarification in addition to discussing with other reviewers.
> - **Hyperparameters.** As mentioned, we discussed all additional explicit hyperparameters introduced by SHLSA in **reply A1 to Reviewer `uX3z`**.
>
>     - For the insensitive parameters $\tau_{\text{pos}}, \tau_{\text{par}}, \tau_{\text{lg}}$, and $\tau_{\text{lw}}$, we adopt an efficient empirical setting approach. Furthermore, we have conducted additional work to avoid explicit parameter specification: **(1)** In **reply A2 to Reviewer `crqa`**, we implemented a ***Confidence-Aware Dynamic Method*** inspired by CACL to adaptively select $\tau_{\text{par}}$, eliminating one explicit parameter. **(2)** We demonstrated the insensitivity of $\tau_{\text{lg}}$ and $\tau_{\text{lw}}$ across different tasks and datasets, showing these parameters require no additional tuning for other fine-grained TTA scenarios. **(3)** For $\tau_{\text{pos}}$, extensive pseudo-labeling literature validates that 0.9 or 0.95 works well for most tasks [R1, R2]. Consequently, four of the five additional parameters can be converted to implicit parameters, leaving only $\tau_{\text{neg}}$ requiring adjustment.
>
>     - For $\tau_{\text{neg}}$, while early training errors pose error accumulation risks, we provided theoretical guidance through Theorem 1 in **reply A3 to Reviewer `crqa`**. Given $\tau_{\text{pos}}$, one only needs to consider error tolerance to select appropriate thresholds (error accumulation rarely occurs when $\epsilon<0.05$). Combined with mitigation strategies from **reply A1[Q1] to Reviewer `uX3z`**, this significantly reduces or eliminates error accumulation impact.
>
>     - In practical applications, insensitive parameters can directly adopt our settings without affecting performance. For $\tau_{\text{neg}}$, Theorem 1 enables threshold calculation with extremely low error bounds (e.g., 0.05), preventing error accumulation. All SHLSA parameters thus require no additional computational overhead and are adaptive across scenarios, effectively becoming **"implicit parameters"** requiring no manual specification.
>
> - **Novelty.** The main contribution of SHLSA, beyond stepwise alignment, is achieving alignment between source and target domains from a **High-Level Semantic** perspective, which **has been acknowledged by Reviewers `iT4e`, `uX3z` and `crqa`**.
>     - Current *Entropy-Based Partition* and *Pseudo-Source Construction* methods introduce *pseudo-source domains*, which is a common characteristic of *Source Distribution Estimation TTA* approaches [R3]. However, they only consider *pseudo-source* and target domains (i.e., ***assuming pseudo-source equivalence to source domain, transforming it into Unsupervised Domain Adaptation***) [R4,R5,R6], enabling models to learn from easy to difficult through curriculum learning and similar methods [R7].
>
>     - SHLSA approaches this semantically, recognizing that **the *pseudo-source domain* still differs from the source domain** and cannot directly serve as the source for UDA tasks. Its core value lies in acting as a **"semantic bridge"** connecting source and target domains. Therefore, we first utilize **"universal semantics"** for calibration, then leverage the calibrated *pseudo-source semantic* to guide samples with larger shifts in the *remaining target domain*. This differs from existing methods using identical training objectives to sequentially adapt *pseudo-source* and target domains (data scheduling), but **truly treats them as different semantic roles with distinct functions**.
>
> Finally, we sincerely thank you for your feedback. We hope the above two clarifications can address your concerns.
>
> [R1] Saito, Kuniaki, et al.. "Asymmetric tri-training for unsupervised domain adaptation." International conference on machine learning. 2017.
>
> [R2] Hu, Yihao, et al. "Selective Label Enhancement Learning for Test-Time Adaptation." The Thirteenth International Conference on Learning Representations.
>
> [R3] Liang, Jian, et al. "A comprehensive survey on test-time adaptation under distribution shifts." International Journal of Computer Vision. 2025.
>
> [R4] Liang, Jian, et al. "Source data-absent unsupervised domain adaptation through hypothesis transfer and labeling transfer." IEEE Transactions on Pattern Analysis and Machine Intelligence. 2021.
>
> [R5] Ding, Yuhe, et al. "Proxymix: Proxy-based mixup training with label refinery for source-free domain adaptation." Neural Networks. 2023.
>
> [R6] Yang, Shiqi, et al. "Casting a BAIT for offline and online source-free domain adaptation." Computer Vision and Image Understanding. 2023.
>
> [R7] Wang, Yuxi, et al. "A curriculum-style self-training approach for source-free semantic segmentation." IEEE Transactions on Pattern Analysis and Machine Intelligence. 2024.

---

### Official Review · Reviewer_iT4e · 2025-11-02

**Soundness:** 3
**Presentation:** 3
**Contribution:** 3
**Rating:** 6
**Confidence:** 4

**Summary:**

The paper introduces a new test-time adaptation method when presented only with a pre-trained source model but no source data and unlabeled target data. To address the limitations of prior work which only look at low-level alignment, the authors present a multi-step alignment approach where they (i) propose a hierarchical feature extractor to fuse local and global signals for structured prediction, (ii) positive and negative psuedo-labels for curriculum learning (iii) stepwise alignment with entropy-based partitioning. The experiment results on both pixel level and image-level prediction problems show the strong gains bought by the method.

**Strengths:**

1. The paper addresses a very important and practical problem of source-data free domain adaptation with an effective solution of step-wise alignment.

2. The idea of stepwise alignment by first selecting a pseudo-source domain in the absence of source data is novel and might have wider use-cases beyond test-time adaptation.

**Weaknesses:**

1. The paper fails to position or compare against other source-free adaptation works such as [1,2,3] which also share the assumption of no source data with only source-trained model. The authors need to include these in the comparisons, or explain why these might not be relevant.
2. For single-image classification setting, the authors also need to include a more challenging dataset such as DomainNet for a better understanding the method's scalability to larger datasets.



[1] Liu, Yuang, Wei Zhang, and Jun Wang. "Source-free domain adaptation for semantic segmentation." Proceedings of the IEEE/CVF conference on computer vision and pattern recognition. 2021.

[2] Kundu, Jogendra Nath, et al. "Generalize then adapt: Source-free domain adaptive semantic segmentation." Proceedings of the IEEE/CVF international conference on computer vision. 2021.

[3] Mitsuzumi, Yu, Akisato Kimura, and Hisashi Kashima. "Understanding and improving source-free domain adaptation from a theoretical perspective." Proceedings of the IEEE/CVF conference on computer vision and pattern recognition. 2024.

**Questions:**

Questions stated above.

---

> ### Author Response · Authors · 2025-11-20
> **Response to Reviewer iT4e (1/1)**
>
> Thank you for taking the time to review our paper and providing valuable feedback. In response to your concerns, we would like to provide the following explanations.
>
> 1. **W1**: Insufficient comparison with source-free adaptation methods.
>
>     **A1**: The three SFDA works [1,2,3] you mentioned are indeed highly relevant to our work, as they all address "adaptation under distribution shift scenarios using only source models and unlabeled target domain data." We have supplemented comparative experiments in **Appendix A.4.2 and A.6.2** of the revised version.
>
>     - **Methodological comparison** (Following the taxonomy on TTDA algorithms from survey [R1])
>         - [1] primarily employs Entropy minimization and Pseudo-labeling strategies, achieving low-density separation through entropy minimization and combining self-training with pseudo-label generation for semantic segmentation adaptation.
>         - [2] combines Consistency Training and Clustering-based Training through a two-stage strategy that first enhances model consistency and then performs target domain clustering optimization.
>         - [3] theoretically unifies Source Distribution Estimation and Consistency Training methods, providing improved virtual domain alignment strategies.
>         - In contrast, SHLSA integrates *pseudo-source domain* generation from Source Distribution Estimation, multi-level label integration from Ensemble-based pseudo labels, and semantic consistency constraints from Consistency Training to achieve progressive adaptation in high-level semantic space.
>
>     - **Experimental details and effectiveness**
>         - [1,2,3] mostly use CNN-based backbones (ResNet-50/101 and VGG-16) and are primarily offline training methods without online testing on streaming data in real scenarios. Additionally, [1,2] mainly target semantic segmentation scenarios.
>         - In comparison:
>             - SHLSA employs different architectures for different tasks, so our comparative experiments mostly target methods with the same backbone, such as [R2, R3]. In the revised version, we will supplement comparative analysis with more methods under different backbones, as shown in **Table R1** entries (o)(p) and **Table R2** entry (m).
>             - Through existing and supplementary experiments, SHLSA demonstrates superior performance in both offline (Table 1, Table 2) and online (refer to the **Reviewer `GWNG`'s reply A5, Table R7**) testing scenarios.
>             - SHLSA analyzes contributions under TTA scenarios with different semantic density levels, primarily testing on semantically dense tasks where it shows more significant advantages. It also validates method generality through single-image classification, a more mainstream TTA scenario.
>
>
> 2. **W2**: Limited evaluation on large-scale datasets in single-image classification setting.
>
>     **A2**: We have supplemented experiments for SHLSA on DomainNet under single-image classification settings. Currently, we include four datasets of varying sizes and challenges: Office-31 (4.6K, 31 categories), Office-Home (15K, 65 categories), VisDA-C (152K, 12 categories), and DomainNet (600K, 345 categories). Results are shown in Table R4:
>
>     **Table R4: Single-Image classification performance of SHLSA on DomainNet.**
>
>     | Method | C$\to$P | C$\to$R | C$\to$S | P$\to$C | P$\to$R | P$\to$S | R$\to$C | R$\to$P | R$\to$S | S$\to$C | S$\to$P | S$\to$R | Avg. |
>     |-|-|-|-|-|-|-|-|-|-|-|-|-|-|
>     |SHOT  |63.5|78.2|59.5|67.9|81.3|61.7|67.7|67.6|57.8|70.2|64.0|78.0|68.1|
>     |ProDe |79.3|91.0|75.3|80.0|**90.9**|75.6|80.4|78.9|75.4|**80.4**|79.2|91.0|81.5|
>     |**SHLSA**|**81.2**|**92.3**|**77.1**|**83.2**|**90.9**|**77.4**|**82.3**|**79.2**|**77.3**|80.1|**82.3**|**93.4**|**83.1**|
>
>     By supplementing the larger-scale DomainNet dataset and combining it with the existing three datasets, we **additionally discovered that SHLSA exhibits different performance gains across data with varying semantic complexity.** Moreover, with increasing numbers of images and categories, it demonstrates scaling effects, confirming the viewpoint that "SHLSA is more suitable for semantically dense tasks." Thank you very much for your suggestion! We have updated this section in **Appendix A.7** of the revised version, marked in brown. It was also mentioned in the **Common Response**, **reply A4 of Reviewer `uX3z`** and the **reply A2 of Reviewer `crqa`**.
>
> [R1] Liang, Jian, Ran He, and Tieniu Tan. "A comprehensive survey on test-time adaptation under distribution shifts." International Journal of Computer Vision. 2025.
>
> [R2] Wang, Yuxi, et al. "A curriculum-style self-training approach for source-free semantic segmentation." IEEE Transactions on Pattern Analysis and Machine Intelligence. 2024.
>
> [R3] Tang, Song, et al. "Proxy denoising for source-free domain adaptation." International Conference on Representation Learning. 2025.

---

> > ### Author Response · Authors · 2025-11-26
> > **Look forward to further discussion. Thank you!**
> >
> > Dear Reviewer iT4e,
> >
> > We hope that our replies and experiments address your concern and we are happy for any further discussion. Look forward to your reply.

---

### Author Response · Authors · 2025-11-20
**Common Response: SHLSA Computational Efficiency Analysis (1/3)**

We sincerely thank all reviewers for taking the time to review our paper and providing valuable feedback. We note that **the computational performance of SHLSA** has received primary attention (`GWNG`, `uX3z`, `crqa`). Here, **1.** we will first conduct a qualitative analysis based on the method's framework, **2.** then perform quantitative experiments with different architectures, **3.** and finally propose and validate feasible performance optimization solutions.

Additionally, since our method and the baseline as well as other comparative methods are generally built on the same architecture (`iT4e`, `GWNG`), we will also supplement the performance results of other related works in part two. The following contents are supplemented in **Appendix A.6** of the revised version and marked in blue.

## **1. Qualitative Analysis of Factors Affecting SHLSA Computational Performance.**

We analyze the performance factors based on the SHLSA framework (SDA+HFA+CACL). The following time complexity analysis describes the changes introduced by SHLSA, and did not fully consider all the details.

First of all, since CACL is a post-processing method requiring only $O(n\log n)$ sorting and $O(n)$ gradient calculation, it introduces minimal computational overhead. Secondly, The computational overhead of SDA stems primarily from using HFA as a feature extractor. As shown in Figure 1(c), *pseudo-source domain* samples undergo HFA twice: once before Mix for semantic correction and once after Mix for semantic guidance. Thus, the main computational increases come from MixMatch operations and our designed HFA.

- **MixMatch Operation:** The time complexity depends on data augmentation times $K$, labeled data (*pseudo-source domain*) batch size $L$, and unlabeled data (*remaining target domain*) batch size $U$, yielding $O(K \cdot L \cdot U)$.

- **HFA Analysis:** Compared to other feature extractors, HFA **(a)** uses local-global information encoding across abstraction levels, and **(b)** performs hierarchical feature fusion via attention modules.

    - **(a) Feature Encoder:** Similar to MobileViT's local attention, this reduces complexity from $O(n^2)$ to $O(k \cdot (\frac{n}{k})^2)$. HFA additionally computes global attention for coarse-grained semantics by resizing the original image to local size before computation, resulting in: $$O((k+1) \cdot (\frac{n}{k})^2) = O(\frac{k+1}{k^2} \cdot n^2).$$ Since $k \in [4, 9]$ in practice, we have $O(\frac{k+1}{k^2} \cdot n^2) < O(n^2)$, so the hierarchical operation itself doesn't increase complexity significantly.

    - **(b) Attention Module:** This performs fusion across $k+1$ patches for $(\frac{n}{k})^2$ pixels, with complexity: $$O((\frac{n}{k})^2 \cdot (k+1)^2) = O(\frac{(k+1)^2}{k^2} \cdot n^2).$$ For semantic segmentation, this is the main overhead source. **For single-image classification, the complexity reduces to $O((k+1)^2)$, causing minimal overhead.**



    - **Overall HFA Complexity:** Combining the analysis of both components, the overall time complexity of HFA is: $$O(\frac{k+1}{k^2}\cdot n^2 + \frac{(k+1)^2}{k^2}\cdot n^2) = O(\frac{(k+1)(k+2)}{k^2} \cdot n^2) > O(n^2).$$ The computational overhead primarily increases during attention fusion of multi-level semantic features in semantic segmentation tasks.

---

> ### Author Response · Authors · 2025-11-20
> **Common Response: SHLSA Computational Efficiency Analysis (2/3)**
>
> ## **2. Quantitative Experiments on the Impact Degree of Different Factors.**
>
> We provide quantitative experimental evidence to support the above qualitative analysis. Specifically, we conduct: **(1)** ablation analysis of computational overhead for different modules based on current SHLSA, **(2)** performance tests of SHLSA under different architectures compared to other methods using the same architectures, and **(3)** evaluations of lightweight TTA methods. We primarily assess GPU Memory (GB), inference time per sample (Time-Item), training time (Time-Train), and final accuracy metrics (mIoU/mAcc).
>
> All experiments are conducted on a single NVIDIA GeForce RTX 3090 (24GB) GPU. For semantic segmentation tasks, we use Synthia$\to$Cityscapes as test data with batch size 2, local window size $\tau_{\text{lw}}$ set to $1024\times1024$, local-global ratio $\tau_{\text{lg}}$ set to 0.5, iterations set to 40,000, and other parameters $\tau_{\text{neg}}, \tau_{\text{pos}}, \tau_{\text{par}}$ as in the paper. For single-image classification tasks, we use Office-Home as test data with batch size 64, $\tau_{\text{lw}}$ set to $256\times256$, epochs set to 10, and other parameters consistent with semantic segmentation tasks. Experimental results are shown in Tables R1 and R2.
>
> **Table R1: Performance and Computational Cost Comparison Across Different Methods on Semantic Segmentation.**
> |Method|GPU Memory (GB)|Time-Item (s/item)|Time-Train (GPU·h)|mIoU (%)|Backbone|Model Size (MB)|
> |-|-|-|-|-|-|-|
> |(a) SHLSA|22.4|2.40|26.7|64.1|Segformer-B5|313.1|
> |(b) SHLSA w/o CACL|22.4|2.40|26.4|63.4|Segformer-B5|313.1|
> |(c) SHLSA w/o HFA|20.2|1.52|18.8|61.3|Segformer-B5|313.1|
> |(d) SHLSA w/o SDA|19.9|2.40|14.3|62.7|Segformer-B5|313.1|
> |(e) SHLSA mit-b4|20.4|2.12|22.1|63.5|Segformer-B4|234.4|
> |(f) SHLSA mit-b3|17.0|1.84|17.4|62.3|Segformer-B3|170.3|
> |**(g) SHLSA mit-b2**|**13.1**|**1.66**|**12.4**|**61.2**|**Segformer-B2**|**94.4**|
> |(h) SHLSA mit-b1|12.6|1.50|11.7|54.3|Segformer-B1|52.2|
> |(i) SHLSA mit-b0|11.7|1.47|11.2|48.9|Segformer-B0|13.7|
> |(j) SHLSA p2t|12.3|1.49|11.5|60.2|P2T-Base|144.5|
> |(k) SHLSA resnet|13.2|1.55|11.3|58.2|ResNet-101|178.9|
> |(l) ATP[R1]|20.1|1.56|19.6|63.7|Segformer-B5|313.1|
> |(m) ATP resnet|12.5|1.51|11.0|57.6|ResNet-101|178.9|
> |(n) ATP p2t|11.9|1.32|9.7|59.6|P2T-Base|144.5|
> |(o) SFKT[R2]|10.8|1.25|8.3|45.9|ResNet-101|178.9|
> |(p) SFDA-Seg[R3]|11.3|1.27|8.5|48.9|ResNet-101|178.9|
>
> **Table R2: Performance and Computational Cost Comparison Across Different Methods on Single-Image Classification.**
> |Method|GPU Memory (GB)|Time-Item (s/item)|Time-Train (GPU·h)|mAcc (%)|Backbone|Model Size (MB)|
> |-|-|-|-|-|-|-|
> |(a) SHLSA|10.8|0.65|0.17|85.0|ResNet-50|102.6|
> |(b) SHLSA w/o CACL|10.8|0.65|0.16|84.9|ResNet-50|102.6|
> |(c) SHLSA w/o HFA|9.7|0.57|0.15|84.6|ResNet-50|102.6|
> |**(d) SHLSA w/o SDA**|**8.2**|**0.65**|**0.13**|**84.7**|**ResNet-50**|**102.6**|
> |(e) SHLSA rn101|13.6|0.67|0.18|89.3|ResNet-101|178.9|
> |(f) SHLSA vgg16|13.9|0.67|0.18|86.7|VGG16|533.4|
> |(g) SHOT[R4]|7.6|0.52|0.12|71.8|ResNet-50|102.6|
> |(h) SHOT rn101|10.4|0.55|0.14|78.9|ResNet-101|178.9|
> |(i) SHOT vgg16|10.6|0.56|0.15|76.3|VGG16|533.4|
> |(j) TENT[R5]|6.9|0.49|0.11|61.7|ResNet-50|102.6|
> |(k) TENT rn101|9.7|0.52|0.13|70.2|ResNet-101|178.9|
> |(l) TENT vgg16|10.2|0.54|0.14|68.3|VGG16|533.4|
> |(m) SFDA+[R6]|7.9|0.55|0.14|73.4|ResNet-50|102.6|
>
> **Analysis Notes:**
> - SHLSA is our proposed method, while SHLSA w/o CACL/HFA/SDA are variants excluding CACL/HFA/SDA during training.
> - CACL is a post-processing operation during training and doesn't affect inference; HFA involves model architecture changes affecting both inference and training (direct backbone input is the general operation without HFA); SDA is a training method affecting only the training process (one-time alignment is the general operation without SDA).
> - For more details, see **reply A4 to Reviewer `GWNG`**. Since CNN-based architectures excel in single-image classification, we focus on CNN architectures without ViT comparisons.
> - Comparing SHLSA's performance across tasks and datasets shows its enhancement scales with increasing semantic complexity. Detailed discussions are in **Common Response (3/3) (Table R3)**, **Reviewer `iT4e`'s reply A2 (Table R4)**, **Reviewer `uX3z`'s reply A4 (Table R8)**, and **Reviewer `crqa`'s reply A2 (Tables R10, R11, R12)**.
>
> **Key Findings from Semantic Segmentation Analysis:**
> - Comparing (a) and (b) in Table R1 shows CACL has minimal impact on training performance; Comparing (a), (b), and (c) demonstrates HFA's significant impact on SHLSA; Comparing (a), (b), and (d) shows MixMatch has moderate impact on SHLSA.
> - Beyond Segformer-B5, we tested various ViT-based architectures (Segformer-B4/B3/B2/B1/B0, P2T) and CNN-based ResNet-101. Comparisons between (a)/(l) and (j)/(n), as well as lightweight methods (o) SFKT/(p) SFDA-Seg versus (k), demonstrate that SHLSA significantly improves segmentation performance with moderate computational overhead increase.

---

> > ### Author Response · Authors · 2025-11-20
> > **Common Response: SHLSA Computational Efficiency Analysis (3/3)**
> >
> > ## **3. Feasible Optimization Solutions.**
> >
> > Based on our analysis, we propose optimization strategies from two perspectives: **Lightweight Attention Mechanisms** and **Simplifying SDA's Two-Step Alignment**.
> >
> > - **Lightweight Attention Mechanisms:**
> >     - The attention module in HFA is the main cause of increased training costs for SHLSA in semantic segmentation scenarios. Following Reviewer `crqa`'s suggestion, we replace the standard attention mechanism with the more efficient MobileViT.
> >     - Results in Table R3 rows (a)(b) and (d)(e) demonstrate that MobileViT achieves comparable performance while reducing GPU memory usage and training time.
> >
> > - **Simplifying SDA's Two-Step Alignment:**
> >     - The stepwise alignment requires *pseudo-source domain* samples to pass through HFA twice for pre- and post-mixing feature extraction. In scenarios with small domain shifts or sparse semantics, directly aligning *pseudo-source domain* and *remaining target domain* (removing SDA) significantly improves computational performance with minimal impact on accuracy.
> >     - For datasets with lower self-entropy values (Office-31: 0.1335, VisDA-C: 0.1677, Office-Home: 0.2491, DomainNet: 0.3478), removing SDA shows minimal performance drops of -0.1%, -0.7%, -0.3%, and -1.2% respectively (92.8%$\to$92.7%, 91.3%$\to$90.6%, 85.0%$\to$84.7%, 83.1%$\to$81.9%) while achieving substantial computational savings.
> >     - However, for dense segmentation scenarios with higher self-entropy values (SYNTHIA$\to$Cityscapes: 0.5396, GTA5$\to$Cityscapes: 0.7239), removing SDA causes significant performance degradation of -1.4% and -3.6% respectively (64.1%$\to$62.7%, 69.2%$\to$65.6%).
> >     - Based on these observations, when average entropy is below approximately 0.5, simplifying SDA's two-step alignment can potentially improve training efficiency without significantly affecting performance. Values in parentheses represent average self-entropy, indicating semantic density and domain shift magnitude.
> >
> > - **Additional Optimization Strategies:**
> >     - Similarly, when domain shift is small, HFA operations can also be simplified to achieve higher computational efficiency.
> >     - Furthermore, optimal solutions can be selected by balancing computational cost and performance trade-offs.
> >     - For detailed discussions, please refer to **Reviewer `crqa`'s reply A1** and **Appendix A.6.3, A.6.4** in revised version.
> >
> > **Table R3: SHLSA Performance Optimization Results**
> > |Methods|GPU Memory (GB)|Time-Item (s/item)|Time-Train (GPU·h)|mIoU/mAcc (%)|Task|Dataset|Backbone|
> > |-|-|-|-|-|-|-|-|
> > |(a) SHLSA|22.4|2.40|26.5|64.1|Semantic Segmentation|SYNTHIA$\to$Cityscapes (0.5396)|Segformer-B5|
> > |(b) SHLSA MobileViT|21.2|2.31|24.2|63.9|Semantic Segmentation|SYNTHIA$\to$Cityscapes (0.5396)|Segformer-B5|
> > |(c) SHLSA w/o SDA|19.9|2.40|14.5|62.7|Semantic Segmentation|SYNTHIA$\to$Cityscapes (0.5396)|Segformer-B5|
> > |(d) SHLSA|22.4|2.40|26.7|69.2|Semantic Segmentation|GTA5$\to$Cityscapes (0.7239)|Segformer-B5|
> > |(e) SHLSA MobileViT|21.2|2.31|24.3|68.9|Semantic Segmentation|GTA5$\to$Cityscapes (0.7239)|Segformer-B5|
> > |(f) SHLSA w/o SDA|19.9|2.40|14.3|65.6|Semantic Segmentation|GTA5$\to$Cityscapes (0.7239)|Segformer-B5|
> > |(g) SHLSA|10.8|0.65|0.06|92.8|Image Classification|Office-31 (0.1335)|ResNet-50|
> > |(h) SHLSA w/o SDA|8.2|0.64|0.04|92.7|Image Classification|Office-31 (0.1335)|ResNet-50|
> > |(i) SHLSA|10.8|0.64|0.17|85.0|Image Classification|Office-Home (0.2491)|ResNet-50|
> > |(j) SHLSA w/o SDA|8.2|0.65|0.13|84.7|Image Classification|Office-Home (0.2491)|ResNet-50|
> > |(k) SHLSA|10.8|0.65|1.72|91.3|Image Classification|VisDA-C (0.1677)|ResNet-50|
> > |(l) SHLSA w/o SDA|8.2|0.64|1.36|90.6|Image Classification|VisDA-C (0.1677)|ResNet-50|
> > |(m) SHLSA|10.8|0.65|1.98|83.1|Image Classification|DomainNet (0.3478)|ResNet-50|
> > |(n) SHLSA w/o SDA|8.2|0.65|1.53|81.9|Image Classification|DomainNet (0.3478)|ResNet-50|
> >
> > [R1] Wang, Yuxi, et al. "A curriculum-style self-training approach for source-free semantic segmentation." IEEE Transactions on Pattern Analysis and Machine Intelligence. 2024.
> >
> > [R2] Liu, Yuang, et al. "Source-free domain adaptation for semantic segmentation." Proceedings of the IEEE/CVF conference on computer vision and pattern recognition. 2021.
> >
> > [R3] Kundu, Jogendra Nath, et al. "Generalize then adapt: Source-free domain adaptive semantic segmentation." Proceedings of the IEEE/CVF international conference on computer vision. 2021.
> >
> > [R4] Liang, Jian, et al. "Do we really need to access the source data? source hypothesis transfer for unsupervised domain adaptation." International conference on machine learning. 2020.
> >
> > [R5] Wang, Dequan, et al. "Tent: Fully test-time adaptation by entropy minimization." arXiv preprint arXiv:2006.10726. 2020.
> >
> > [R6] Mitsuzumi, Yu, et al. "Understanding and improving source-free domain adaptation from a theoretical perspective." Proceedings of the IEEE/CVF conference on computer vision and pattern recognition. 2024.

---

### Comment · Area_Chair_Z8CP · 2025-11-26
**Reminder: Discussion Phase Engagement Needed**

Dear Reviewer iT4e, uX3z and crqa:

As the deadline for the discussion phase is approaching in less than one week, could you kindly engage in the discussion with the other reviewers and provide your response to the authors’ rebuttal?

Best regards,

AC

---

### Author Response · Authors · 2025-11-29
**Summary of Paper Contributions and Reviewer Discussions for Quick Overview**

Dear ACs, SACs and PCs,

Thank you for your efforts in maintaining community fairness. We provide here a brief and objective summary of the **(1) paper's main contributions** and our **(2) previous discussions with reviewers** to help you quickly understand the current status of the paper.
## **1. Main Contributions of SHLSA.**
- We propose a **stepwise alignment method from a semantic perspective** for source-free test-time adaptation scenarios with unlabeled target domains under distribution shift.
- Specifically, **(a)** we first utilize entropy to partition the target domain, constructing a *pseudo-source domain* with features closely resembling the source domain and a *remaining target domain* with more significant distributional shifts. **(b)** Subsequently, we leverage the universal semantics of pre-trained models to rectify the *pseudo-source domain*. **(c)** Finally, we employ the rectified *pseudo-source domain* to guide the adaptation of the *remaining target domain*. **(d)** Additionally, we introduce HFA and CACL to enhance the quality of semantic features obtained throughout the alignment process.
- Throughout this process, the *pseudo-source* domain serves as a **semantic bridge**, mitigating the semantic inconsistencies that arise in prior Source Distribution Estimation (SDE) based methods which directly substitute the *pseudo-source domain* for the source domain. Our approach achieves superior performance across diverse downstream tasks, exhibiting especially notable gains on semantically rich tasks including semantic segmentation and multi-label classification.
## **2. Reviewer Feedback and Our Responses.**
> As of Nov 27, reviewer `GWNG` has replied to our rebuttal, stating that the final decision will be determined based on further discussions. The other three reviewers (`iT4e`, `uX3z`, `crqa`) have temporarily not provided their responses. *All the content below can be found in the rebuttal and the revised version already provided.*
- **Strengths**
    - Stepwise semantic alignment in source-free TTA is an effective and novel approach (`iT4e`, `uX3z`, `crqa`).
    - The paper's presentation is clear and reasonable with rigorous logic (`GWNG`, `uX3z`).
    - Provides theoretical support with comprehensive experiments and significant improvements (`iT4e`, `GWNG`, `uX3z`).
    - Provides pseudocode and anonymous repository for easy reproduction (`crqa`).
- **Weaknesses**
    - **Common Issues**:
        - **Additional computational overhead** (`GWNG-[W2]`, `uX3z-[W3]`, `crqa-[W1]`): We provided qualitative analysis and quantitative experiments in the **Common Response** to validate SHLSA's effectiveness and offered optimization solutions.
        - **Marginal improvement on single-label classification tasks** (`uX3z-[W4]`, `crqa-[W2]`): From a semantic density perspective, we supplemented SHLSA's analysis with semantic segmentation and single-label classification experiments, and evaluated performance on the semantically complex DomainNet dataset and multi-label classification tasks, demonstrating that SHLSA's effectiveness increases with task semantic complexity.
        - **Lack of comparison experiments across different architectures and with more methods** (`iT4e-[W1]`, `GWNG-[W6]`, `ux3z-[W2]`): Combined with computational performance analysis, we supplemented comparison experiments using Segformer-B0/B1/B2/B3/B4/B5, P2T, and CNN-based architectures on semantic segmentation tasks, and provided comparisons using ResNet-50/101 and VGG16 architectures on single-label classification. We also included introductions to all comparison methods in the revised version.
        - **Lack of more challenging test data and online testing scenarios** (`iT4e-[W2]`, `GWNG-[W5]`): We supplemented our evaluation with the semantically complex DomainNet dataset and provided experimental settings and results for online scenarios.
        - **Parameters setting and tuning** (`ux3z-[W1]`, `crqa-[W2,W3]`): We analyzed SHLSA parameter sensitivity and provided adaptive setting methods for certain parameters and theoretical guidance for key parameters, beyond the mitigation solutions already presented.
    - **Individual Issues**:
        - **`GWNG-[W1]`** (Generalization limitations of pre-trained semantics): We demonstrated both qualitatively and quantitatively that pre-trained semantics provide universal value, independent of fine-grained semantics.
        - **`GWNG-[W3]`** (Insufficient novelty): We detailed comparisons with existing works, proving SHLSA's innovative contribution in stepwise alignment from a semantic perspective.
        - **`GWNG-[W4]`** (Lack of more detailed ablation experiments): We supplemented analysis of ablation experiments for SHLSA's three modules across all scenarios.
        - **`crqa-[W3]`** (Insufficient description of Theorem 1): We supplemented introductions to all variables in Theorem 1 and proved that the theorem still holds under severe distribution shift conditions.

---

### Meta-Review · Area_Chair_7jRk · 2025-12-28

**Summary:**

iT4e: (1) Lack of comparison with other source-free adaptation works. (2) Need to include more challenging dataset such as DomaiNet.

GWNG: (1) Need to have more analysis on cross-task transferability. (2) Need more detailed runtime or memory comparison. (3) Lack of novelty. (4) Need more thorough analysis on cross-module dependencies. (5) Need more challenging or dynamic scenarios. (6) Need more evaluation across diverse architectures.

uX3z: (1) numerous hyperparameters require tuning. (2) Lack of introduction to baselines. (3) Computation overhead. (4) The performance improvement for image classification is negligible.

crqa: (1) Need to explore potential optimization to reduce computation cost. (2) Need more exploration on the limitations. (3) Disconnect between theory and practice in Theorem 1.

The ratings of this paper are mixed. While the rebuttal addressed some of the reviewers' concerns, the AC believes some concerns are still outstanding. First of all, several reviewers pointed out that the novelty of the paper is small. The AC agrees. Most contributions of the paper are fairly well known (hierarchical feature, confidence-aware learning, etc). The main idea (stepwise alignment) of the paper is essentially making use of the uncertainty of pseudo-labels in the target domain. This idea has been explored in many other source-free domain adaptation work. This paper is basically one way of operationalizing this idea (i.e. grouping data in the target domain with high certainty into a pseudo-source domain, then treat it specially). Second, the use of imagenet feature (GWNG) raises some concerns, since many of the datasets in the experiments (especially for classification) can be considered as subsets of ImageNet. Also the limited performance improvement (uX3z) is a valid concern, since the paper uses fairly complex pipeline compared with other source-free domain adaptation works. Given these concerns, the paper is not ready for ICLR in its current form. Authors are encouraged to take into account of reviewers' comments and revise it for resubmission

**Reviewer Concerns:**

iT4e: the concerns are addressed

GWNG: most concerns are addressed. The reviewer still has outstanding concerns on  hyperparameters and novelty

uX3z: (1)(2)(3) are addressed in the rebuttal. (4) (limited performance improvement) is still outstanding

crqa: most concerns are addressed

**Reviewer Scores:**

One reviewer (GWNG) responded and mentioned that he/she will make the final decision after discussions. Other reviewers have not responded yet. Based on the reviews and rebuttal, the scores are unlikely to change to a positive rating.

---

### Decision · Program_Chairs · 2026-01-26

Reject